# Multi-centennial climate change in a warming world beyond 2100

Sun-Seon Lee[1,2*], Sahil Sharma[1,2], Nan Rosenbloom[3], Keith B. Rodgers[4*], Ji-Eun Kim[1,2], Eun Young Kwon[1,2], Christian L.E. Franzke[1,2], In-Won Kim[1,2], Mohanan Geethalekshmi Sreeush[5], and Karl Stein[1,2]

[1]Center for Climate Physics, Institute for Basic Science, Busan, Republic of Korea
[2]Pusan National University, Busan, Republic of Korea
[3]National Center for Atmospheric Research, Boulder, CO, USA
[4]WPI-Advanced Institute for Marine Ecosystem Change, Tohoku University, Sendai, Japan
[5]Alfred Wegener Institute, Helmholtz Centre for Polar and Marine Research, Am Handelshafen 12, 27570 Bremerhaven, Germany

*Correspondence to*: Sun-Seon Lee (sunseonlee@pusan.ac.kr) and Keith B. Rodgers (keith.rodgers.b2@tohoku.ac.jp)

**Abstract.** Changes in the climate due to human influences are expected to extend well beyond the 21st century. Despite growing interest in climate change after 2100 and improved computational resources, multi-century climate projections remain limited in number. Here, we examine a set of 10 ensemble simulations extending the Community Earth System Model 2 large ensemble (CESM2-LE) from 2101 to 2500 under the shared socio-economic pathway (SSP)3-7.0 scenario, which involves the reduction of fossil and industrial $CO_2$ emissions to zero by 2250. By the year 2500, substantial forced changes are projected in both the spatial and temporal characteristics of variability and mean states. Post-2100, El Niño-Southern Oscillation variability is expected to diminish, while the tropical intraseasonal variability will notably strengthen. Global mean surface temperature and precipitation are projected to continue rising even after $CO_2$ emissions cease. Additionally, substantial soil carbon release from permafrost thawing is projected over Siberia and Canada, resulting in a shift of land from a carbon sink to a carbon source after the 22nd century. The ocean experiences a rapidly diminished capacity to absorb anthropogenic $CO_2$ after the 21st century, while nevertheless continuing to act as a carbon sink, with an increased contribution from the Southern Ocean to total carbon uptake. The model also projects a considerable decline in low-latitude marine primary production, which is linked to a considerable depletion of dissolved inorganic phosphate in the local mesopelagic domain. The extended simulations predict substantial changes in the amplitude and timing of precipitation seasonality at the urban scale, with variations across different locations. Similarly, seasonal variations in the partial pressure of $CO_2$ in seawater along different latitudinal bands are projected to experience distinct changes. These findings suggest that post-2100 changes will not simply be an extension of the trends projected for the 21st century. Taken together, these new simulations highlight the far-reaching impacts of multi-centennial climate change on both human societies and global ecosystems.

## 1. Introduction

To date, the scientific literature on climate change projections has primarily focused on projecting changes to the year 2100 (e.g., IPCC, 2021), motivated by the need to guide policy and societal decisions that affect multiple human generations. However, several critical components of the climate system such as heat and carbon reservoirs will respond slowly over timescales of multiple centuries, and their response times will play a strong role in determining the impact of the long-term climate change (Oh et al., 2024; Armour et al., 2016; Koven et al., 2022; Schuur et al., 2015; Cheng et al., 2019). Therefore, given that the legacy of human-induced perturbations will persist well beyond 2100, particularly through the multi-century to multi-millennia residence time of $CO_2$ in the atmosphere (Archer et al., 2009), protocols have been developed to extend Coupled Model Intercomparison Project (CMIP) simulations to 2500 (Meinshausen et al., 2020). Very recently, scenarios for long-term extensions up to 2500 have been newly proposed for the ScenarioMIP experiments of CMIP phase 7 (ScenarioMIP-CMIP7) (van Vuuren et al., 2025). The growing interest in these extended timescales can be seen in the increasing number of studies that consider multi-century projections of climate change (e.g., Randerson et al., 2015; Moore et al., 2018; Koven et

al., 2022; Hezel et al., 2014; Mahowald et al., 2017; Wang et al., 2024; Peng et al., 2024; Geng et al., 2024). Under relatively strong emissions scenarios, these studies emphasize that the adverse impacts of climate change are expected to persist well beyond 2100. As an example, an inter-model comparison of global carbon cycle simulations identified general consistencies in the sign of mean state responses to strong anthropogenic forcing, focusing on carbon uptake by the ocean and terrestrial systems that has potential impacts for climate feedbacks (Koven et al., 2022). In the models, the uptake of carbon by the ocean gradually declines, but remains positive throughout the 22nd and 23$^{\text{rd}}$ centuries. The terrestrial carbon cycle tends to shift from a net sink to either a neutral state or a net source.

Given the expected importance of ocean overturning structures in sustaining marine ecosystems (Sarmiento et al., 2004), it is also essential to examine how marine ecosystems may be impacted over longer timescales. This was first considered for the case of an extension simulation with an Earth system model to 2300 by Moore et al. (2018). Under historical and extended Representative Concentration Pathway (RCP) 8.5 forcing to 2300 with the Community Earth System Model 1 (CESM1), the study reported a 24 % decrease in primary production and a 41 % decrease in the carbon export across the 100 m horizon in the low latitudes (30°S-30°N) to 2300. This decrease was argued to be due to enhanced nutrient retention in the Southern Ocean due to a poleward shift of westerlies and sea ice melt, which then in turn led to a reduction in the supply of nutrients to the low latitudes by a Subantarctic Mode Water conduit. The interpretative framework of Moore et al. (2018) built on the long-standing paradigm for Southern Ocean dominance of low-latitude productivity, originally considered for the mean state by Sarmiento et al. (2004).

Subsequently, Rodgers et al. (2024) revisited the question of future changes in low-latitude primary production and export using five distinct CMIP phase 6 (CMIP6) models, arguing instead for the critical role of low-latitude remineralization and water mass renewal in determining the response to sustained warming over multiple centuries. They underscored the pivotal role of the transfer efficiency of the ocean's twilight or mesopelagic zone, defined as the ratio of the export flux at the base of the mesopelagic domain to the export flux at the base of the euphotic zone, in elucidating the transient responses of Earth system in the deep future. Although they established that thermocline retention of nutrients serves as a first-order control on future projections, emphasizing the central importance of temperature-dependent remineralization, they did not address the degree to which this might be modulated by the stoichiometric plasticity mechanism proposed by Kwon et al. (2022) and Tanioka et al. (2022), both of which had evaluated simulations spanning from 1850 to 2100.

In the context of forced changes in variance within the Earth system, recent studies have explored the sustained anthropogenic impacts on the variability of the El Niño-Southern Oscillation (ENSO), arguing for a reduction in ENSO variability under strong forcing beyond 2100 (Peng et al., 2024; Geng et al., 2024). Additionally, a recent study presented the possibility of a permanent El Niño with reduced ENSO amplitude in a high-$CO_2$ world, but that study also pointed out that unforced variability at the centennial timescale for this case is comparable to the projected changes under a wide range of $CO_2$ forcing (Callahan et al., 2021). Future ENSO projections including changes in amplitude, frequency, phase-locking, and skewness, particularly the post-2100 ENSO changes, remain highly uncertain. For this reason, further investigation into future ENSO changes on longer-term timescales is needed. One related recent finding is that the Indian Ocean Dipole variability, which interacts with ENSO, is projected to increase in the eastern pole in the 21$^{\text{st}}$ century, followed by a reduction thereafter (Wang et al., 2024). These findings indicate that forced changes in Earth system variability may exhibit non-linear responses to sustained greenhouse warming after 2100. Consequently, to inform long-term adaptation planning and decision-making, it is of great value to have multi-century projections derived from an ensemble of simulations with a single model, which can provide insights into potential risks associated with ongoing greenhouse warming.

Regarding multi-centennial timescales, several stabilization experiments have been conducted, including those with constant atmospheric greenhouse gas concentrations (Dittus et al., 2024; Fabiano et al., 2024) or net-zero $CO_2$ emission simulations (King et al., 2024), to study climate projections under stabilized warming and the dependence on different levels of forcing. However, these experiments are based on a single simulation. For this study, we have chosen to investigate forced changes in the climate system out to the year 2500 by extending 10 members of the 100-member Community Earth System Model 2 large ensemble (CESM2-LE; Rodgers et al., 2021) from 2101 to 2500. This is done by leveraging an extension of the shared socio-economic pathway (SSP)3-7.0 scenario applied to the CESM2-LE. The choice of not only an ensemble of simulations but also the application of CESM2 offers several advantages for understanding future changes beyond 2100. An ensemble of transient simulations provides a more robust characterization of forced changes in the mean state and variability (Deser et al., 2020). In addition, advances in soil biogeochemistry representation with vertical resolution in Community Land Model Version 5 (CLM5; Lawrence et al., 2019) enhance the study of how permafrost thaw could impact the soil carbon reservoir in high latitudes of the Northern Hemisphere after the 21$^{st}$ century.

## 2. Extension of 10 members of CESM2-LE to 2500

Here, we present a 10-member extension of the CESM2-LE to the year 2500. The original CESM2-LE employs historical (1850-2014) and projected (2015-2100) SSP3-7.0 forcing fields (O'Neill et al., 2016), with the latter representing a medium-high reference scenario within the "regional rivalry" socio-economic family (Meinshausen et al., 2020). All CESM2-LE simulations have nominal 1° x 1° spatial resolution. The selected 10 members from the CESM2-LE for extension are the macro-perturbation simulations which were initialized from 10 model states ranging from year 1011 to 1191 (20-year intervals) of a pre-industrial control simulation conducted with CESM2 (Danabasoglu et al., 2020). For a fuller description of CESM2-LE, the reader is referred to Rodgers et al. (2021).

A detailed description of the underlying model components of CESM2 can be found in Danabasoglu et al. (2020). For the purposes of this study, we first summarize the model components before describing the forcing and boundary conditions used in the extended simulations. The atmospheric model component is the Community Atmosphere Model version 6 (CAM6; Danabasoglu et al., 2020), and thus is distinct from the model components from the single ensemble member of CESM2-WACCM simulation to 2300 under SSP5-8.5 presented by Koven et al. (2022). The land model consists of the CLM5 (Lawrence et al., 2019). The physical ocean and sea ice components are the Parallel Ocean Program version 2 (POP2; Smith et al., 2010) and the CICE Version 5.1.2 (CICE5; Bailey et al., 2020), respectively. CESM2 uses the Marine Biogeochemistry Library (MARBL; Long et al., 2021) for ocean ecosystem modeling. In the context of interpreting the sustained strong ocean warming impacts on biogeochemical and ecosystem changes, it is important to note that MARBL does not account for the effect of ocean warming on remineralization rates. As noted by Rodgers et al. (2024), this may have important implications for projected future concentrations of mesopelagic and thermocline nutrients, and consequently for primary production and export.

Meinshausen et al. (2020) provided greenhouse gas concentrations, defined as dry air mole fractions, for both standard and extended SSP scenarios. They used the reduced-complexity climate-carbon-cycle model MAGICC7.0 ('Model for the Assessment of Greenhouse Gas Induced Climate Change') to produce future greenhouse gas concentrations driven by harmonized SSP greenhouse gas emissions (Gidden et al., 2019) and extended emissions beyond 2100. To extend the CESM2-LE from 2101 to 2500, we followed the extended SSP3-7.0 protocol, a concentration-driven configuration. In this extended scenario, fossil and industrial $CO_2$ emissions are effectively ramped down to zero by 2250 (Meinshausen et al., 2020), as shown in Fig. 1a. Figure 1b presents the time evolution of global mean greenhouse gas mole fractions ($CO_2$, $CH_4$, $N_2O$, and CFCs) which are prescribed in these simulations under the historical (1850-2014), standard SSP3-7.0 (2015-2100), and

extended SSP3-7.0 (2101-2500) scenario forcings. The global mean atmospheric $CO_2$ mole fraction at the end of the 25th century, provided by the extended SSP3-7.0 scenario, is approximately 1371 ppm. Even though fossil and industrial $CO_2$ emissions decrease after the year 2100 and are ramped down to zero by 2250 in the simulations, the atmospheric $CO_2$ concentration (i.e., mole fraction) peaks around the year 2240 (at approximately 1515 ppm) and declines modestly thereafter. The top-of-the-atmosphere radiation imbalance exhibits a strong increase throughout the 21st century, peaking at approximately

3.5 W m$^{-2}$ in the mid-22nd century, and then gradually decreasing to approximately 1.6 W m$^{-2}$ by the year 2500 (Fig. 1c). Under the SSP3-7.0 forcing, global mean surface air temperature perturbations exceed 12 °C at the end of 25th century relative to the 1850-2019 period, i.e., the HadCRUT4 (Morice et al., 2012) observation period (Fig. 1d). Note that the CESM2 has a very high climate sensitivity (Gettelman et al., 2019). If the 1850-1900 baseline is used, the global mean surface air temperature change closely matches the change observed with the 1850-2019 baseline.

Additionally, SSP3-7.0 is characterized by relatively strong land use changes, particularly a global reduction in forest cover by 2100, along with high emissions of near-term climate forcers including tropospheric aerosols, tropospheric $O_3$ precursors, and $CH_4$ (O'Neill et al., 2016). Among the SSP extension scenarios, SSP3-7.0 is the scenario with the highest atmospheric concentrations for both $CH_4$ and $N_2O$ (Meinshausen et al., 2020). All other forcings, including aerosols and land use, were fixed at the values of the year 2100 in the extended simulations. This study uses 10 ensemble simulations from both CESM2-

LE and extended simulations covering the entire period from 1850 to 2500 (651 years).

We also incorporated output from the RAD simulation, known as radiatively coupled in CMIP terms, to understand carbon cycle perturbations. In the RAD simulations, a constant pre-industrial $CO_2$ concentration is maintained for biogeochemical fluxes of $CO_2$ over land and ocean, e.g., air-sea gas exchange, while the change in atmospheric $CO_2$ affects the radiation balance of the atmosphere only.

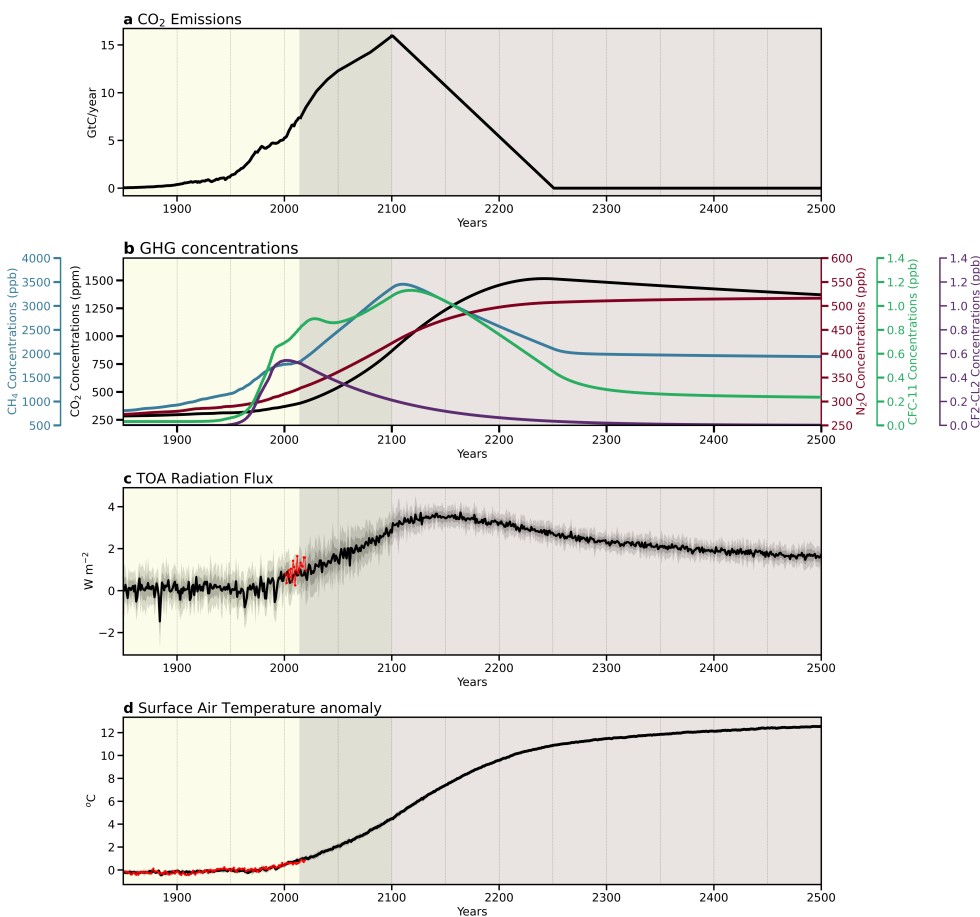

**Figure 1: Time series of global mean (a) fossil fuel and industrial $CO_2$ emissions and (b) greenhouse gas mole fractions over 1850-2500 for the CESM2-LE extension simulations. Values are taken from Meinshausen et al. (2020). Time series of global fields over**

1850-2500 for 10 ensemble members for (c) top-of-atmosphere radiative imbalance (W m$^{-2}$) along with the CERES-EBAF product (red) (Loeb et al., 2018; Loeb et al., 2009) and (d) anomalies of global mean surface air temperature (°C) along with HadCRUT4 (red) (Morice et al., 2012). In (c) and (d), bold lines represent ensemble means, and dark and light shading represent 1 standard deviation (SD) and 2 SD variability. In (d), observed and simulated temperature anomalies are calculated with respect to the period spanned by the observations (period 1950-2019).

## 3. Results

### 3.1 Multi-centennial changes in mean state and variability

To provide a perspective on the overall long-term climate changes, in addition to the changes in the top-of-the-atmosphere radiation imbalance and temperature (Fig. 1c, d), we begin by presenting the evolution of mean state for several key variables over the time interval from 1850 to 2500 (Fig. 2). The anomalies shown in Fig. 2 were calculated relative to the observation period for each case that includes an observational dataset, following the procedure used in the CESM2-LE overview study (Rodgers et al., 2021). The global mean precipitation anomalies show gradual increases from the 21$^{st}$ century to the end of the 25$^{th}$ century, with a 23.5 % increase in global mean precipitation over the 2401-2500 period relative to the 1979-2020 satellite observational period (Fig. 2a). Overall, annual mean precipitation increases are evident across all latitudinal bands, except for 45°S-15°S and 15°N-40°N (Fig. S1a-f).

Ocean heat accumulation, as indicated by changes in ocean heat content, provides a measure of the excess anthropogenic heat stored in the ocean (Garcia-Soto et al., 2021), and complements global mean surface temperature changes. The observed changes indicate that global ocean heat content in the upper 2,000 m increased by 351.4 ± 59.8 ZJ from 1958 to 2019 (Cheng et al., 2022). Our simulations also show a consistent global ocean warming in the upper 2,000 m throughout the entire simulation period (Fig. 2b). By the end of the 25$^{th}$ century, the projections indicate a cumulative increase of ~17,000 ZJ, which is approximately seven times higher than the projected heat content perturbation by the end of the 21$^{st}$ century in the CESM2-LE (Rodgers et al., 2021). Based on the idealized and comprehensive overshoot simulations, it is noted that human-induced ocean warming and deoxygenation are altering marine ecosystems, potentially resulting in a centuries-long, irreversible loss of habitable ocean volume in the upper 1000 m (Santana-Falcón et al., 2023). A rapid growth in ocean heat content leads to considerable changes in sea ice melting, exerting a large influence on the global heat balance. In contrast to the stabilized Arctic sea ice extent projected by around 2040 under the low-emission SSP1-2.6 scenario of CMIP6 (Davy and Outten, 2020), our simulations project a continuous decline of Arctic sea ice from the late 20$^{th}$ century to the mid-22$^{nd}$ century, followed by an ice-free condition thereafter (Fig. 2c). A similar trend pattern is projected for the Southern Ocean sea ice as well. The loss of sea ice and resultant environmental changes at a local scale have already influenced weather and climate outside of the Arctic and Antarctic regions (e.g., Screen et al., 2018). Given the crucial role of the Atlantic Meridional Overturning Circulation (AMOC) in distributing heat within the Earth's climate system, a number of previous studies have evaluated projected AMOC changes (e.g., Ditlevsen and Ditlevsen, 2023; Weijer et al., 2020; Liu et al.; Roberts et al., 2020). As shown in many climate model simulations including CESM2-LE (e.g., Rodgers et al., 2021; Weijer et al., 2020), our extended simulations project a slowing of the AMOC in response to anthropogenic warming until the late 22$^{nd}$ century, with a slight recovery following the minimum state of the AMOC strength (Fig. 2d). However, it is noted that in the idealized experiments of Zero Emission Commitment Model Intercomparison Project (ZECMIP), some models indicate AMOC strengthening, while others predict a continued decline after 50 years of CO$_2$ emissions cessation, illustrating the model dependency of future AMOC projections (MacDougall et al., 2022).

The response of net primary productivity (NPP) in the terrestrial biosphere and the ocean to anthropogenic forcing is an indicator of changes in the rate at which carbon is fixed at the base of respective food webs over the land and ocean, with implications for global ecosystems. While marine NPP shows a relatively modest decrease after the end of the 21$^{st}$ century, terrestrial NPP exhibits a considerable increase from the end of the 20$^{th}$ century to the mid-22$^{nd}$ century and it saturates

thereafter (Fig. 2e). The fact that changes are much larger in terrestrial NPP than in marine NPP is due to terrestrial NPP being strongly modulated by changing atmospheric $CO_2$ concentrations through $CO_2$ fertilization effects (e.g., Prentice et al., 2001). Our extended simulations suggest that the globally integrated ocean NPP declines by 12 % from 48 PgC yr$^{-1}$ in 1850 to 42 PgC yr$^{-1}$ in 2500. As we will see in Figs. 9-11, future low-latitude decreases in marine NPP are accompanied by large decreases in $PO_4$ and $NO_3$ throughout the ocean's mesopelagic domain (100 m to 1000 m depth), with this having a deleterious impact on the rate at which macronutrients become available within the euphotic zone through ocean circulation and mixing. The uptake rates of $CO_2$ by global land and ocean increase as anthropogenic $CO_2$ emissions increase until the end of the 21$^{st}$ century (Fig. 2f). However, after the end of the 21$^{st}$ century, the uptake rates of both land and ocean are projected to decrease, indicating reductions in net terrestrial and oceanic carbon sinks. The substantial decline in oceanic carbon uptake, with positive values indicating a flux into the ocean, can be described as a weakened solubility pump through a reduced $CO_2$ buffering capacity of seawater, as well as through a reduction in the soft tissue component of the biological pump and a strong reduction in AMOC overturning (e.g., Chikamoto and DiNezio, 2021). Nevertheless, our simulations indicate that global ocean persists as a weakened carbon sink until the year 2500. The decrease in net carbon uptake rate of the terrestrial system after the end of the 21$^{st}$ century is largely attributed to an increase in microbial carbon decomposition and the release of carbon in regions impacted by permafrost thawing. In our simulations, the permafrost regions, defined as areas where the active layer thickness of soils is less than 3 m, begin to decline at the end of the 20$^{th}$ century, followed by a continued reduction until the mid-22$^{nd}$ century (Fig. 2g). Future changes in terrestrial and marine biogeochemistry, as well as potential climate-carbon feedbacks, will be discussed further in Section 3.3.

Our simulations demonstrate increases in greenhouse warming-driven stability in both the atmosphere and the ocean over global scales. A positive anomaly in atmospheric stability from the early 21$^{st}$ century onwards indicates increased stabilization of the atmosphere relative to its present-day climate conditions (Fig. 2h). The upper tropospheric amplification, i.e., enhanced warming of the upper troposphere relative to the surface (e.g., Santer et al., 2017), is responsible for this increased stability of the atmosphere. Also, an increase in the ocean's surface temperature due to greenhouse warming leads to an enhanced upper ocean stratification, as measured by the density differences between the 300 m and 50 m depths. The stronger upper ocean stratification is associated with a decrease in the mixed layer depth (Fig. 2i, j).

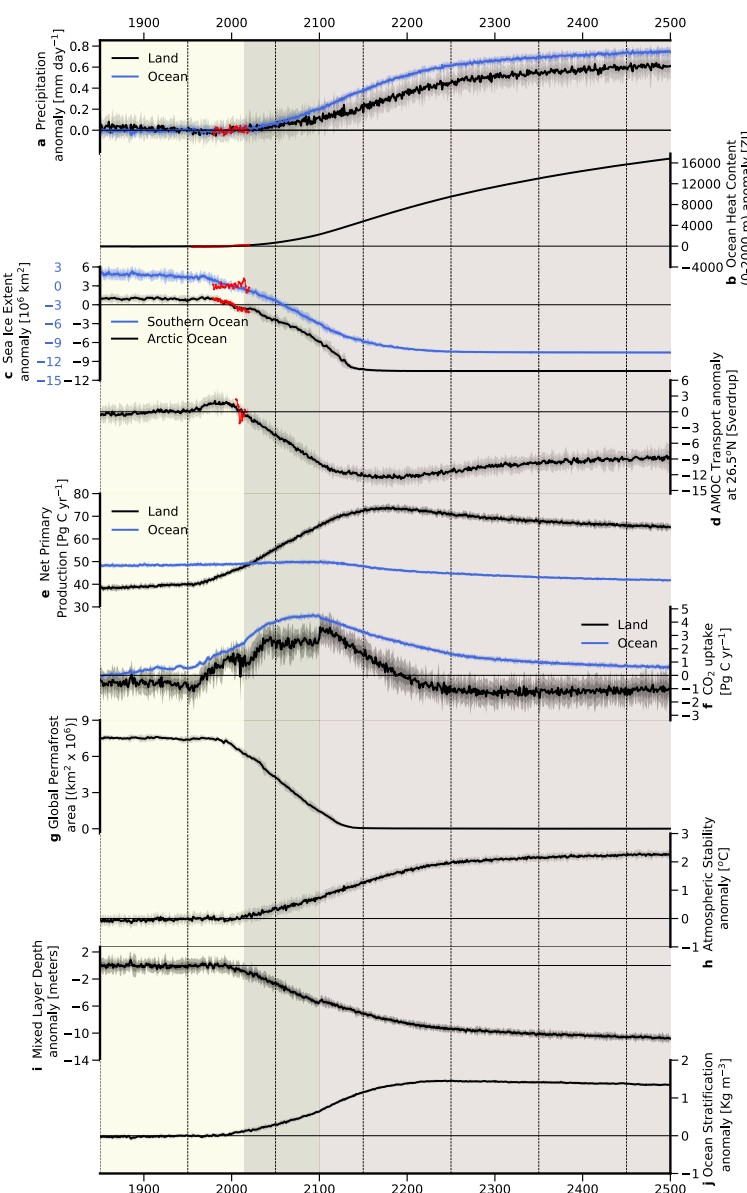

**Figure 2: Time series of global fields over 1850-2500 for 10 ensemble members.** (a) Anomalies of the precipitation (mm day⁻¹) over the land (black) and over the ocean (blue) along with the Global Precipitation Climatology Project (GPCP) (Adler et al., 2012; Adler et al., 2003), (b) anomalies of ocean heat content integrated over the upper 2000 m (ZJ) along with an observation-based product (Ishii et al., 2017), (c) anomalies of sea ice extent (10⁶ km²) for the Arctic (black) and Southern Ocean (blue), with observed sea ice extent over 1979-2020 (Fetterer, 2017), (d) Atlantic Meridional Overturning Circulation (AMOC) transport anomalies at 26.5°N (Sv), with RAPID array observations (Frajka-Williams et al., 2019), (e) globally integrated net primary productivity (NPP) (Pg C yr⁻¹) over the land (black) and over the ocean (blue), (f) globally integrated net $CO_2$ uptake (Pg C yr⁻¹) over the land (black) and over the ocean (blue), (g) global permafrost area defined as the active layer thickness is less than 3 m (km² x 10⁶), (h) globally averaged atmospheric stability (difference between 200 hPa air temperature and surface air temperature) anomaly (°C) relative to 1850-1949 period, (i) anomalies of mixed layer depth (m) relative to 1850-1949 period, and (j) ocean stratification (difference between potential density at 300 m and 50 m) anomaly (kg m⁻³) relative to 1850-1949 period. For model fields, bold lines represent ensemble means, and dark and light shading represent 1 standard deviation (SD) and 2 SD variability. For each case, where observational products (red) are included, anomalies are calculated with respect to the period spanned by the observations. All fields considered here are annually averaged values. For comparison with the 100-member CESM2-LE over 1850-2100, the reader is referred to Fig. 1 of Rodgers et al. (2021), where a subset of the variables shown here are presented.

Figure 3 displays the spatial distributions of climatologies over the 1850-1949 period and projected future changes for several key fields. Here, we consider the periods 2150-2249 and 2401-2500 as the 'extended future' and the 'deep future' periods, respectively. With a global mean warming level greater than 12 °C by the end of the 25th century relative to the 1850-2019 observation period (Fig. 1d), more pronounced warming over land is projected for the extended future period, and the warming is then intensified in the deep future (Fig. 3a-c). The strongest warming, exceeding 28 °C, is found over Nunavut, Canada. High-latitude regions are projected to warm more substantially than low-latitude regions (Fig. 3c), except for the central and eastern equatorial Pacific, implying the continuation of polar amplification (e.g., Koven et al., 2022) and thereby as a

consequence a weakened equator-to-pole surface temperature gradient beyond 2100, in particular over the Northern Hemisphere. Regarding changes in the tropical Pacific mean state, most climate models show a reduction in their zonal sea surface temperature (SST) gradient due to preferentially stronger eastern relative to western equatorial Pacific warming under increased $CO_2$ concentrations (e.g., Vecchi and Soden, 2007; Fredriksen et al., 2020). By the 25th century, the eastern equatorial Pacific shows an enhanced warming of 8-10 ℃ relative to the 1850-1949 period, leading to a further weakening of the west-

east SST gradient beyond 2100. This change affects ocean-atmosphere interactions encompassing the Pacific Walker circulation (e.g., Chung et al., 2019; Zhang and Karnauskas, 2017), tropical precipitation structure (e.g., Chadwick et al., 2013), ENSO variability (e.g., Cai et al., 2021; Fredriksen et al., 2020; Peng et al., 2024), and ultimately global ecosystems (Lee et al., 2022).

The spatial distribution of precipitation changes for the 25th century relative to the historical period is similar to the pattern of

changes by the end of the 21st century shown in Rodgers et al. (2021). However, the magnitude of perturbations is substantially larger (Fig. 3f). One notable feature of the tropical precipitation changes in our extended simulations is the retreat or disappearance of the climatological South Pacific Convergence Zone (SPCZ), which is characterized by a northward shift with loss of its diagonal orientation towards the subtropics relative to the historical climatology (Fig. S2b, c). A comparison between 50-year periods (1950-1999 and 2050-2099) reveals that several CMIP5 and CMIP6 models show a northward shift of the

SPCZ in the future, consistent with our model projections, while others predict a southward shift (Narsey et al., 2022). The equatorward movement and the projected change in the zonal structure of the SPCZ in the future can be linked to the SST response in the central equatorial Pacific (Cai et al., 2012). Additionally, our simulations project a southward shift of the climatological Intertropical Convergence Zone (ITCZ), as part of the coupled climate system response to the AMOC slowdown as indicated by previous studies (e.g., Bellomo et al., 2021; Vellinga and Wood, 2008; Zhang and Delworth, 2005), resulting

in substantial precipitation increase along the equator and the merging of two tropical precipitation bands. As a result, tropical precipitation zones are projected to become narrower (Lau and Kim, 2015), and the dry regions between the two tropical convergence zones are likely to experience wetter conditions in the post-2100 (Fig. S2a-c, Fig. 3d-f). The projected precipitation changes over the southern mid-latitudes and the Southern Ocean are expected to be linked to both Southern Ocean warming and the resulting meridional temperature gradient reduction between the tropics and the Southern Ocean (Grose and

King, 2023).

Meanwhile, the increased precipitation along the equatorial Pacific is accompanied by greater upward atmospheric motion in this region (Fig. 3h, i and Fig. S2e, f). More broadly, the enhanced anomalous upward motion is accompanied by weakened easterlies over the tropical Pacific Ocean (Fig. 3k, l), which is likely linked to the changes in the east-west SST gradient (Fig. 3c), suggesting a weakening of the Pacific Walker circulation due to greenhouse warming. Our simulations project an overall

weakening of the easterlies in the Pacific and Atlantic, spanning approximately 30°S to 30°N, with a substantial reduction along the equator in the Pacific in the extended future. This also reflects the weakening of Walker circulation. These changes are projected to become even more pronounced in the deep future (Fig. 3k, l). In contrast to the changes in the tropics, westerly winds in the Southern Hemisphere show an intensification poleward of 50°S with a reduction equatorward of 50°S (Fig. 3l), consistent with a poleward shift of westerlies found in previous studies (e.g., Deng et al., 2022).


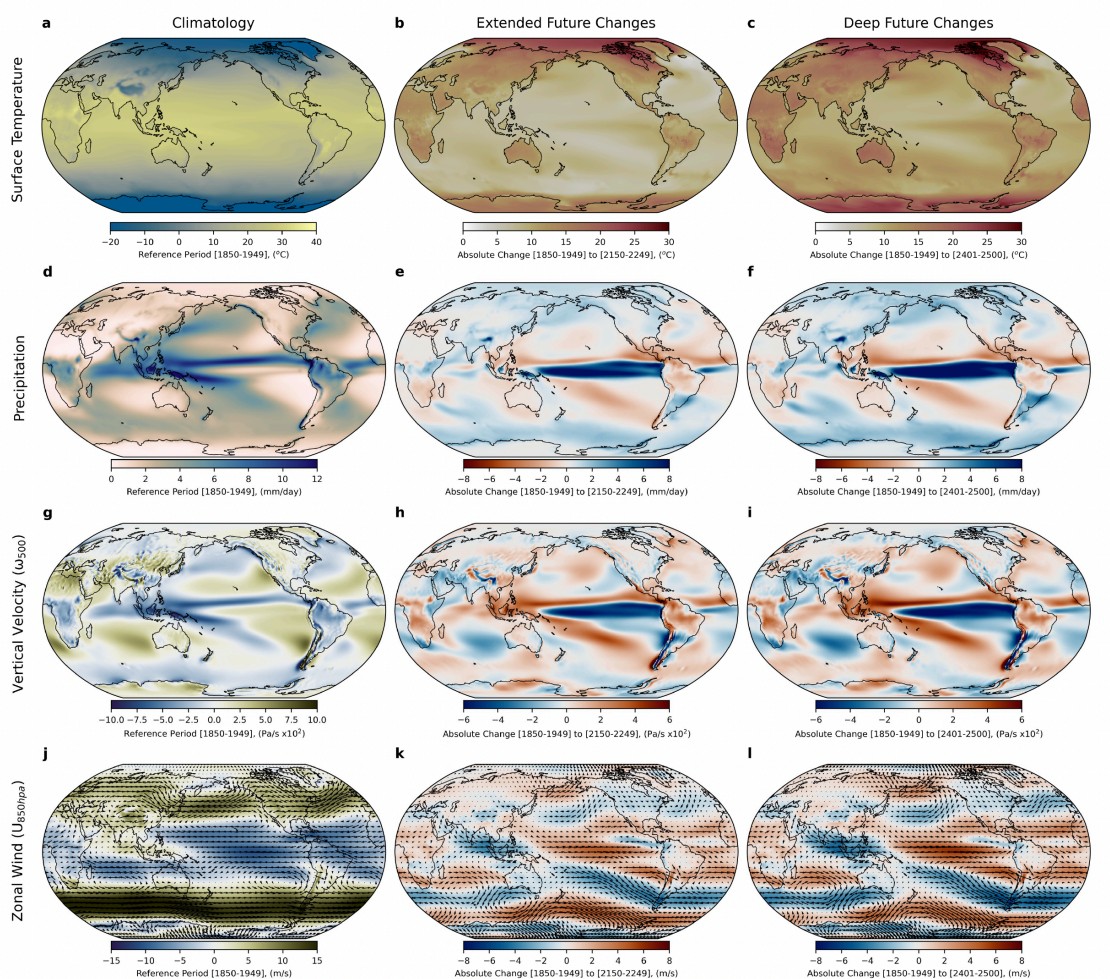

**Figure 3: Historical climatology (period 1850-1949) (first column) and future perturbations over the period 2150-2249 (second column) and over the period 2401-2500 (third column) relative to 1850-1949. (a-c) Surface temperature (°C), (d-f) precipitation (mm day⁻¹), (g-i) 500 hPa vertical p-velocity (Pa s⁻¹ x 10²), and (j-l) 850 hPa wind (vector, m s⁻¹) and its zonal component (shading, m s⁻¹).**


In addition to examining spatial patterns of mean state changes, we also investigated changes in the spatial pattern of surface temperature and precipitation variability (Fig. 4). Except for Africa, south/southeast Asia, Australia, South America, and the North Atlantic regions, the inter-ensemble spread in surface air temperature is projected to decrease, especially over high-latitude land regions in the Northern Hemisphere (Fig. 4b, c), indicating strong forced changes in variance over these regions.

Similarly, suppressed temperature variability is projected over the equatorial Pacific, suggesting a reduction in ENSO variability. The variability of ENSO, calculated by the 30-year running standard deviations of monthly Niño3.4 SST, exhibits a substantial reduction beyond the 21$^{st}$ century (Fig. 4d), which is consistent with previous studies (Peng et al., 2024; Geng et al., 2024). These studies suggest that reduced ENSO variability is linked to a weakened thermocline feedback due to collapsed equatorial ocean upwelling and the spread of convection to the eastern equatorial Pacific. Unlike surface temperature, the

dominant change in future precipitation variability consists of an increase over most regions, with the exception of the tropical zone within the latitude bands spanning approximately 5°N-25°N and 35°S-10°S, where a decline in the mean precipitation is projected. As a consequence of the equatorward shift and merging of two tropical precipitation bands (Fig. 3e, f), precipitation variability along the equator in the Pacific is projected to increase strongly by the deep future period (Fig. 4g). Precipitation variability in the Niño3.4 region shows a peak in the mid-21$^{st}$ century along with ENSO SST variability, but there is a rebound

in the mid-23$^{rd}$ century (Fig. 4h).

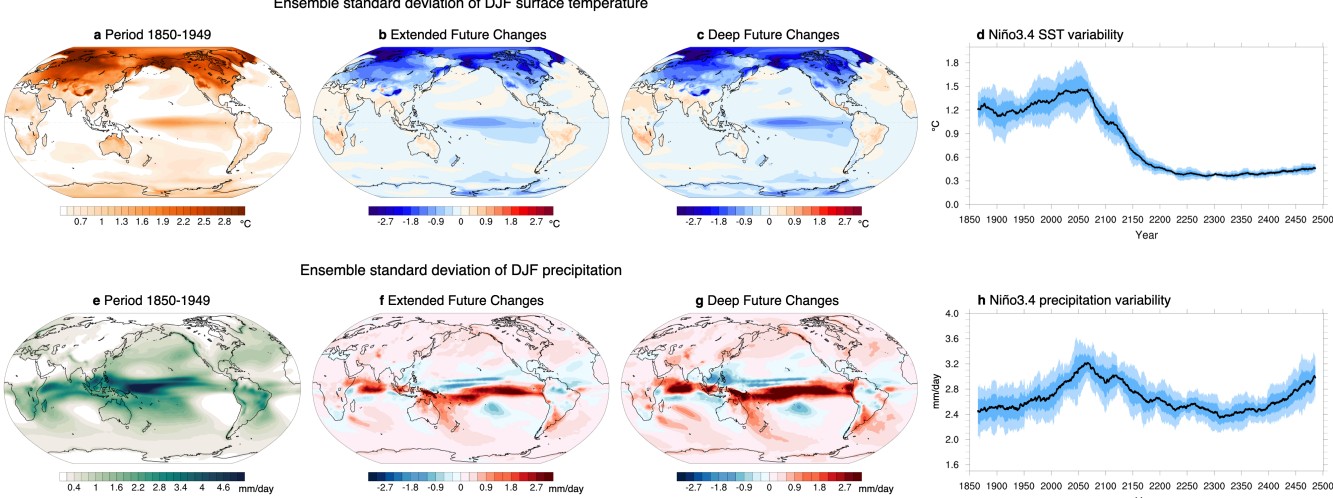

Figure 4: Time-averaged across-ensemble standard deviation of the boreal winter (DJF) mean (a) surface temperature (°C) and (e) precipitation (mm day$^{-1}$) over the period 1850-1949. Future changes of the standard deviation values of (b, c) surface temperature and (f, g) precipitation over the period 2150-2249 and over the period 2401-2500, respectively, relative to 1850-1949. These calculations involve first determining the standard deviation across all ensemble members for the same time period, followed by averaging across time. (d, h) 30-year running standard deviations of monthly Niño3.4 SST and precipitation, respectively.

Furthermore, we explored future changes in the Madden-Julian Oscillation (MJO), the dominant mode of tropical intraseasonal variability during boreal winter (Madden and Julian, 1972). Due to minor adjustments to the deep convection scheme and the inclusion of the unified turbulence scheme, MJO representation in CESM2 has improved relative to CESM1 (Danabasoglu et al., 2020). As a result, the MJO's eastward movement from the Western Indian Ocean into the central Pacific is represented in CESM2 (Danabasoglu et al., 2020). Here, overall MJO activity is assessed through a wavenumber-frequency diagram (Wheeler and Kiladis, 1999; Waliser et al., 2009) with 10°S-10°N averaged daily precipitation and 850 hPa zonal wind for the historical and future periods (Fig. 5). During the historical period, for both precipitation and 850 hPa zonal wind, maximum power is observed in the 30-80 day period and wavenumber 1-2, with greater eastward than westward power (Waliser et al., 2009). In the extended and deep future periods, the spectral amplitude in the intraseasonal frequency domain is projected to increase markedly, indicating a potential strengthening of the MJO intensity in response to greenhouse warming (Figs. 5b, c, e and f). Furthermore, the maximum variance of both precipitation and 850 hPa zonal wind is projected to shift towards higher frequencies, with the maximum occurring at periods shorter than 30 days, closer to 25 days (Cui and Li, 2019). In particular, it is projected that the amplitude of power spectrum for precipitation in the extended and deep future periods will be approximately ten times greater than those observed in the historical period (Fig. 5c). Thus, in these extended simulations, we identify a remarkable future transition from a case where ENSO variability is dominant in the equatorial region to a state where it is greatly diminished and the MJO becomes more prominent. This shift could have a number of implications for local ecosystems and biogeochemical cycling in the tropics, as well as for teleconnections away from the regions impacted directly. Although our study provides valuable insights, the results should be interpreted with caution due to the limitations of using output from a single Earth system model to represent long-term changes in climate variability. Therefore, further investigations utilizing simulations from multiple Earth system models would be highly beneficial in identifying areas of general consistency across models, as well as potential areas of substantial disagreement.

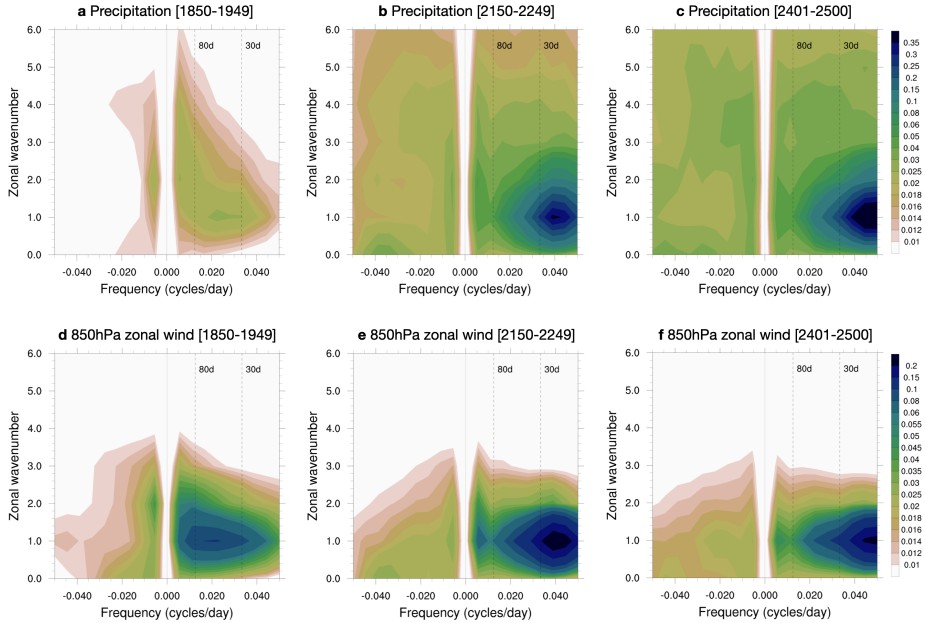

**Figure 5: Wavenumber-frequency power spectra for 10°S-10°N averaged precipitation (mm² day⁻²; upper panels) and 850 hPa zonal wind (m² s⁻²; lower panels) for November-April. (a, d) Period 1850-1949, (b, e) period 2150-2249, and (c, f) period 2401-2500. The spectra are calculated for individual ensemble members and then averaged across the 10 ensemble members.**

## 3.2 Changes in seasonality

As described in the previous section, sustained multi-century warming trends will alter the spatial and temporal characteristics of mean precipitation. Additionally, changes in the seasonal phasing and the magnitude of precipitation at a regional scale will drive substantial hydrological changes along with impacts on the water resources. For instance, when averaged over the tropical band between 15°S and 15°N, the seasonal cycle of precipitation in the 24th and 25th centuries is projected to have two peaks, occurring in May and August, instead of the single peak that is typically observed from mid-June to mid-July over the historical period (Fig. S1c, i). Furthermore, mean state changes are expected to lead to shifts in frequency, intensity, and duration of precipitation extremes at the local scale (Sun et al., 2022), potentially resulting in substantial social and economic losses. Figure S1 indicates that day-to-day variations in spatially averaged precipitation tend to increase along with the increases in time averages. This is especially evident to the north of 40°N and south of 45°S during rainy seasons, implicating future increases in the magnitude and frequency of extreme precipitation over those regions. Therefore, a better understanding of the signature of forced changes at the regional or local scales can provide valuable insights into how strong anthropogenic forcing will impact extreme precipitation events and water availability.

Figure 6 shows the distributions of the seasonal cycles of precipitation and extreme precipitation distributions over several megacities across disparate geographical regions. In general terms, a megacity designates a city with a population of more than 10 million people. In 2018, there were 33 megacities in the world, accounting for 7 % of the world's total population, and the number of megacities is projected to increase to 43 by the year 2030 (UN-DESA, 2018). In the present study, we have selected seven megacities located in diverse geographical regions (Table 1) to examine the forced response of precipitation seasonality and extreme precipitation changes at a local scale for illustrative purposes. Our simulations project substantial changes in precipitation seasonality in terms of its phase and strength. In Tokyo (Japan), which was the world's most populous city in 2018, the phase of precipitation seasonality is projected to remain relatively stable from the historical period through the 25th century. However, the magnitudes of the two peaks are expected to increase in the post-2100 period (Fig. 6 and Fig. S3a). Meanwhile, in São Paulo (Brazil), the wet season in the current climate state spans from October through March (Silva Dias et al., 2013), but in the future, June-October are projected to become the months which have the largest precipitation, with a

notable increase in precipitation magnitude from the 23rd century onward (Fig. 6 and Fig. S3b). A similar phase shift is projected to occur in Moscow (Russia) which is currently the northernmost megacity in 2018. Our extended simulations indicate that after the mid-22nd century, summer precipitation will decrease, while winter precipitation is projected to double relative to the historical period (i.e., 1850-1949). As a result, the patterns of wet summers and dry winters will be reversed (Fig. 6 and Fig. S3g). New York (United States) also indicates a strong increase in future winter precipitation (Fig. 6 and Fig. S3e). In addition to these cities depicted in Fig. 6, more generally, our extended simulations project that the Northern Hemisphere high-latitude regions will experience strong increases in winter mean precipitation (Fig. S1f).

The analysis also identifies delayed peaks in the seasonal cycle for some megacities. From the historical period to the 25th century, a peak shift from July to August is apparent for Mumbai (India), and it is accompanied by a nearly two-fold increase in the maximum monthly precipitation (Fig. 6 and Fig. S3d). Our simulations suggest that sustained anthropogenic warming could bring drier conditions to Mexico City (Mexico), with delayed peaks in July and October during the 25th century, compared to peaks in June and September over the period 1850-1949 (Fig. 6 and Fig. S3c). Lagos (Nigeria) experiences, in general, a wet season from April to October and a dry season from November to March in the present-day climate (Guo et al., 2022). In the extended future, the precipitation peak in July is projected to shift to September, and this peak will be weakened after the 23rd century with an overall decreasing trend (Fig. 6 and Fig. S3f). Our results point to the heterogeneity of the shifts in the seasonal cycle of precipitation across different regions, highlighting the diverse impacts of Earth's changing climate at the regional-to-urban scales.

Although there are differences in the shape and magnitude of precipitation seasonality changes among the selected megacities, future changes in extreme precipitation exhibit a consistent trend across all locations (Fig. 6). In other words, the likelihood of extreme daily precipitation will increase in the future with sustained greenhouse warming (Giorgi et al., 2019; Gründemann et al., 2022). For instance, in Tokyo, the probability of heavy precipitation exceeding 250 mm day$^{-1}$ is projected to increase by an order of magnitude. As the seasonal cycle of precipitation and frequency of extreme events can directly and indirectly impact human activities, water resource management, and the availability of food, further studies on precipitation seasonality and identifying hot spots of weather extremes, supported by additional model simulations, are essential.

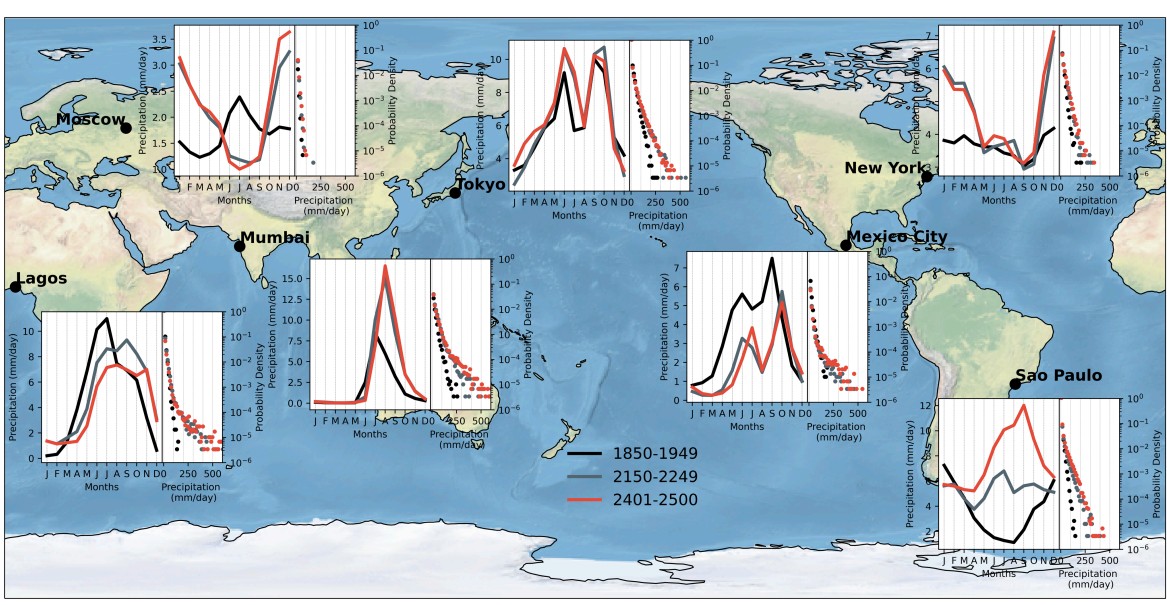

**Figure 6: Seasonal cycle of monthly precipitation (solid lines, mm day$^{-1}$) and probability density function of daily precipitation (dots, %) for seven megacities (Tokyo, São Paulo, Mexico City, Mumbai, New York, Lagos, and Moscow) over the period 1850-1949 (black), 2150-2249 (gray), and 2401-2500 (red). For more details on the seven selected megacities, see Table 1. The background image is shown using the Natural Earth dataset included with the Cartopy mapping library in Python.**

**Table 1 Megacities (total population above 10 million) that are selected in the present study.**

| City | Country | Region | Population (UN-DESA, 2018) |
|---|---|---|---|
| Tokyo | Japan | East Asia | 37,468,000 |
| São Paulo | Brazil | South America | 21,650,000 |
| Mexico City | Mexico | North America | 21,581,000 |
| Mumbai | India | South Asia | 19,980,000 |
| New York | United States | North America | 18,819,000 |
| Lagos | Nigeria | West Africa | 13,463,000 |
| Moscow | Russia | Europe | 12,410,000 |

Another key variable that is projected to undergo fundamental changes in seasonality is the surface ocean partial pressure of $CO_2$ ($pCO_2$). Although there is disagreement regarding the amplitude and phase of the seasonal variations in $pCO_2$ between the climate models and observation-based products, which can be attributed largely to differences in the seasonal variability

of the surface dissolved inorganic carbon (DIC) concentrations (Rodgers et al., 2023), an increasing trend in the amplitude of the seasonal cycle of $pCO_2$ has been addressed in several studies (e.g., Joos et al., 2023; Rodgers et al., 2008). It is also noted that the amplitude of global ocean $pCO_2$ seasonality is expected to intensify by a factor of 1.5 to 3 over the 2080-2100 period relative to 2006-2026 (Gallego et al., 2018). However, it remains unclear how the $pCO_2$ seasonality may change beyond 2100. From the historical period to the 25[th] century across all latitude bands, we identify an intensification of the $pCO_2$ seasonal cycle

amplitude (i.e., maximum minus minimum) (Fig. 7), except for the tropics. The future amplification in $pCO_2$ seasonality can be attributed to the increased sensitivity of ocean $pCO_2$ to changes in DIC and temperature contributions (e.g., Gallego et al., 2018), or equivalently to the invasion flux of anthropogenic carbon amplifying the thermally-driven component of the seasonal cycle in $pCO_2$ (Fassbender et al., 2022). Seasonal variability of $pCO_2$ in the tropics is small during the historical period, consistent with observations and CMIP6 simulations (Joos et al., 2023), and the tropical ocean still shows weak seasonality

beyond 2100 (Fig. 7c). North of 40°N, the timing of maxima and minima in the projected seasonal cycle in the post-2100 period differs notably from the seasonality in the historical period, showing an earlier occurrence of seasonal minimum and maximum oceanic $pCO_2$ around April and July, respectively. In the Southern Ocean, the peak in March-April after the end of the 21[st] century can be attributed to the increasing influence of temperature over DIC (Gallego et al., 2018).

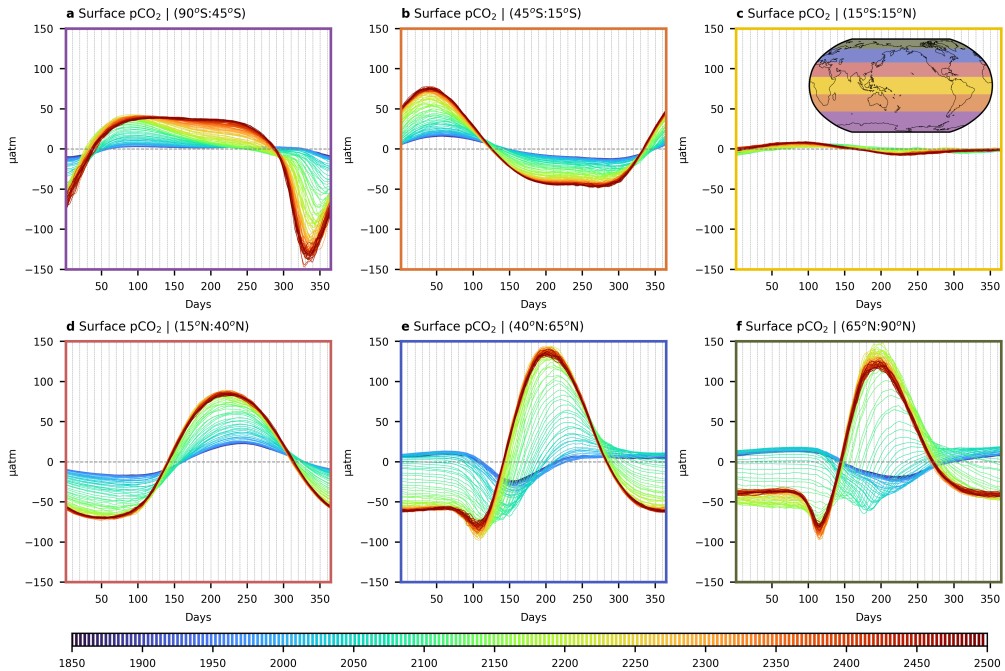

**Figure 7: Seasonal cycle of ocean $p$CO$_2$ ($\mu$atm) along latitudinal bands in 5-year intervals. (a) 90°S-45°S, (b) 45°S-15°S, (c) 15°S-15°N, (d) 15°N-40°N, (e) 40°N-65°N, and (f) 65°N-90°N.**

**3.3 Changes in terrestrial and marine ecosystems and potential carbon-climate feedbacks**

Several studies have previously considered carbon-climate feedbacks based on model simulations and observations (Friedlingstein and Prentice, 2010; Koven et al., 2022; Friedlingstein et al., 2006; MacDougall et al., 2020), but uncertainty in quantifying climate-carbon cycle feedbacks in the post-2100 period is still large. Here, we consider broadly future changes in terrestrial and marine ecosystems, as well as the potential carbon-climate feedbacks, based on our extended simulations. Figure 8 illustrates the temporal and spatial changes in total terrestrial carbon stored in soils and vegetation, which can play a vital role in the terrestrial biosphere. The global terrestrial carbon stocks increase from 856 PgC in 2000 to 2288 PgC in 2200, and then decrease to 2079 PgC by 2500 (Fig. 8a). The spatial map shows a notable decrease in terrestrial carbon in parts of the subarctic and Arctic regions, central Africa, and southern Brazil, while showing an increase in the Amazon, Indonesia, parts of Asia, the United States, and southern Canada during the 25th century relative to 1850-1949 (Fig. 8g). The decreasing trend in terrestrial carbon after the 22nd century is largely driven by changes in soil carbon stocks (Fig. 8b, h). A substantial loss of soil carbon is projected to occur after the mid-22nd century, which is consistent with the timing of permafrost disappearance (Fig. 2g). Soils in the permafrost regions contain enormous amounts of organic carbon (Fig. 8e), which has the potential to be released into the atmosphere as carbon dioxide and methane (e.g., Schuur et al., 2015). Even though there may be a strong positive feedback of permafrost carbon to the global climate (Natali et al., 2021), atmospheric radiation computations in our model use prescribed atmospheric CO$_2$ concentrations, and thereby carbon released from thawing soil in permafrost regions is not feeding back into the atmospheric radiation budget. This is because we follow the CMIP6 convention of following a concentration - rather than an emission - forcing for our simulations. Based on a coupled model of intermediate complexity which includes a fully coupled representation of oceanic and terrestrial carbon cycles, it is estimated that carbon-climate feedbacks could result in an additional warming of 0.13-1.69 °C by 2300, which is independent of the pathway of anthropogenic emissions (MacDougall et al., 2012). In the deep future period, global soil carbon continues to decline at a similar rate, implying a cumulative additional warming that could be twice as large as the current estimated range of 0.13-1.69 °C by the year 2500. Changes in the soil carbon in the deep future over the tropical and mid-latitude bands are relatively small compared to the changes over the Arctic and subarctic regions (Fig. S4a-c). The global vegetation carbon stock increases from

494 PgC in 2000 to 873 PgC in 2200, which can be attributed to CO₂ fertilization effects (Fig. 8c). After reaching 900 PgC in 2250, the vegetation carbon stock declines slightly, consistent with the temporal evolution of atmospheric CO₂ concentrations (Fig. 1b). Meanwhile, in the high and mid-latitudes of the Northern Hemisphere, the vegetation carbon storage increases by a factor of two (Fig. S4d). Notably, vegetation carbon strongly increases in the Amazon and Indonesian rainforests, while a decrease occurs in the African rainforest. The observed saturation and decline of carbon sinks in tropical forests (Koch et al., 2021), along with our findings, would have implications for the potential future contribution of vegetation carbon losses to atmospheric CO₂ levels.

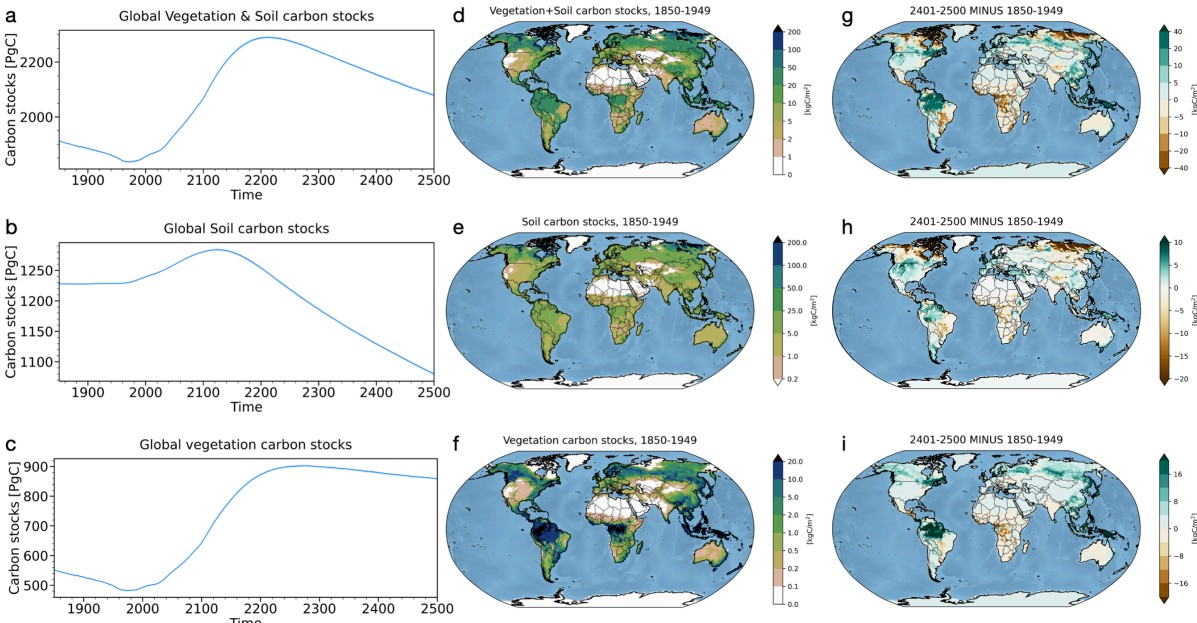

**Figure 8: Global (a) vegetation and soil carbon stocks (PgC), (b) soil carbon stocks (PgC), and (c) vegetation carbon stocks (PgC). Historical climatology (period 1850-1949) for (d) vegetation and soil carbon stocks (kgC m⁻²), (e) soil carbon stocks (kgC m⁻²), and (f) vegetation carbon stocks (kgC m⁻²). Future changes for (g) vegetation and soil carbon stocks (kgC m⁻²), (h) soil carbon stocks (kgC m⁻²), and (i) vegetation carbon stocks (kgC m⁻²) for the period 2401-2500 relative to 1850-1949.**

Next, we revisit the approximately 15 % drop in globally integrated marine NPP shown in Fig. 2e. As a substantial fraction of newly produced organic matter is vertically exported below the surface and remineralized into inorganic matter in the ocean's interior - a process known as the biological pump - NPP changes within the euphotic zone propagate into changes in particulate organic carbon (POC) fluxes throughout the water column. In other words, export represents a net downward transport of organic material (typically expressed in carbon mass units) across a depth horizon, with this component of the flux being primarily sustained through gravitational sinking. Primary production is the rate at which organic material is produced from inorganic compounds and does not have a directional component. Given the predominance of recycling within the sunlit surface layers of the ocean, primary production is almost always larger than export. Regionally, ocean NPP increases by up to ~80 % in the Southern Ocean and decreases by up to 50 % in the mid-latitudes of the North Atlantic in the extended future (Fig. 9b). The spatial patterns of NPP change are qualitatively similar to the changes in POC export at a depth of 100 m (Fig. 9e), and the regional increases and decreases in each of NPP and POC export tend to amplify towards the 25[th] century (Fig. 9c, f), in response to persistent changes in ocean overturning and associated stratification changes. Over the polar regions, large reductions in sea ice cover can increase light and nutrient availability and lengthen the growing seasons of phytoplankton, both of which result in increasing NPP. Over broad areas of the Southern Ocean, where a pronounced projected increase in NPP is evident, enhanced productivity in the surface ocean is co-located with nutrient trapping within the upwelling-dominated Southern Ocean, further fueling NPP (Moore et al., 2018). In contrast, NPP declines in oligotrophic gyres and in the Northern

Hemisphere mid-latitudes. The largest decrease identified over the North Atlantic can be attributed to reductions in nutrient concentrations associated with AMOC slowing and oligotrophication of the subpolar North Atlantic, along with increased ocean stratification (Fig. 2). The NPP reductions in oligotrophic gyres are also driven by surface nutrient depletion, suggesting that nutrient uptake plasticity, an adaptive strategy used by phytoplankton to reduce nutrient uptake while maintaining carbon fixation (i.e., increasing carbon-to-phosphorus ratios in phytoplankton cells or communities) when surface nutrients are scarce, becomes less effective beyond the 21st century. This strategy, which plays a key role in sustaining ocean NPP until the 21st century (Kwon et al., 2022), loses its effectiveness in later periods.

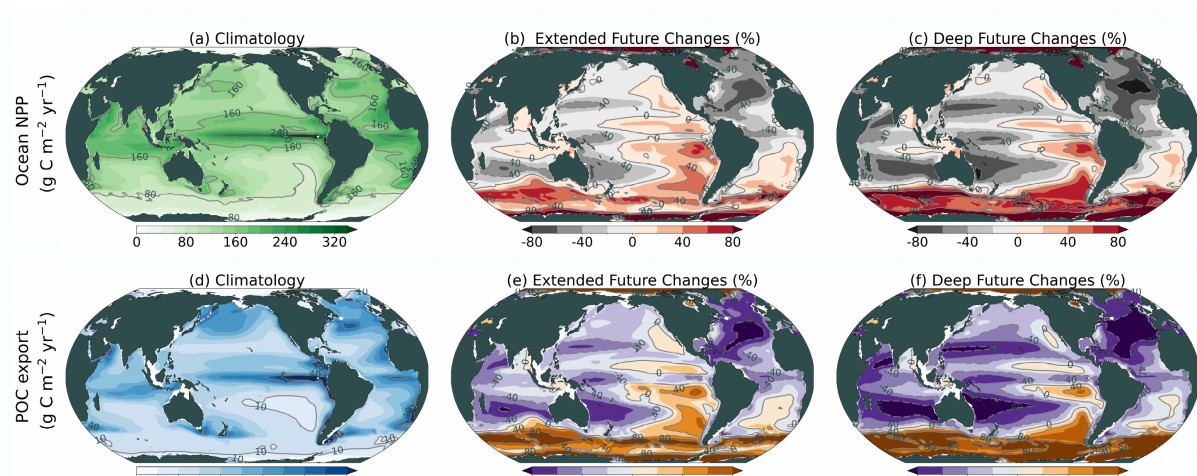

**Figure 9: Historical climatology (period 1850-1949) (first column), and future perturbations over the period 2150-2249 (second column) and over the period 2401-2500 (third column), relative to 1850-1949. (a-c) Ocean NPP (gC m$^{-2}$ yr$^{-1}$) and (d-f) POC export (gC m$^{-2}$ yr$^{-1}$).**

To interpret the changes in projected global biogeochemical cycles shown in Fig. 9, it is instructive to consider the relative projected changes in both dissolved inorganic phosphate ($PO_4$) and dissolved inorganic nitrate ($NO_3$) concentrations integrated over the euphotic zone depth, which we approximate by averaging over the 0-100 m depth interval for each case. We consider the future changes as a ratio between future (averaged over the interval 2150-2249 or 2401-2500) and historical (averaged over the interval 1850-1949) periods. Our simulations reveal broad decreases in $PO_4$ in the extended future, with the largest reductions throughout the subtropical regions (Fig. 10a). Projected changes in $NO_3$ contrast with the changes in $PO_4$, as $NO_3$ shows a patchwork of both increases and decreases across different global regions (Fig. 10c). By the 25th century, $PO_4$ concentrations in the euphotic zone are depleted to very low levels throughout the subtropics, as well as across the subpolar North Atlantic and Arctic (Fig. 10b). In contrast, $NO_3$ exhibits an even more pronounced pattern of strong increases and decreases in concentrations across different regions (Fig. 10d). These results indicate that, over the full latitudinal range spanned by the shallow overturning subtropical cells, approximately 45°S-40°N, there will be a transition to $PO_4$-limitation for primary production under future climate change. Unlike $PO_4$, a future increase in dissolved inorganic iron (Fe) is evident, except in high-latitude regions of both hemispheres (Fig. 10e, f). The spatial patterns of dissolved inorganic silicate ($SiO_3$) changes are similar to those of $NO_3$, featuring a weak increase in low and mid latitudes and a strong decrease in high-latitude regions (Fig. 10g, h). The increases in $NO_3$, Fe, and $SiO_3$ in low to mid latitudes may result from reduced uptake rates by phytoplankton under severe P-limited conditions.

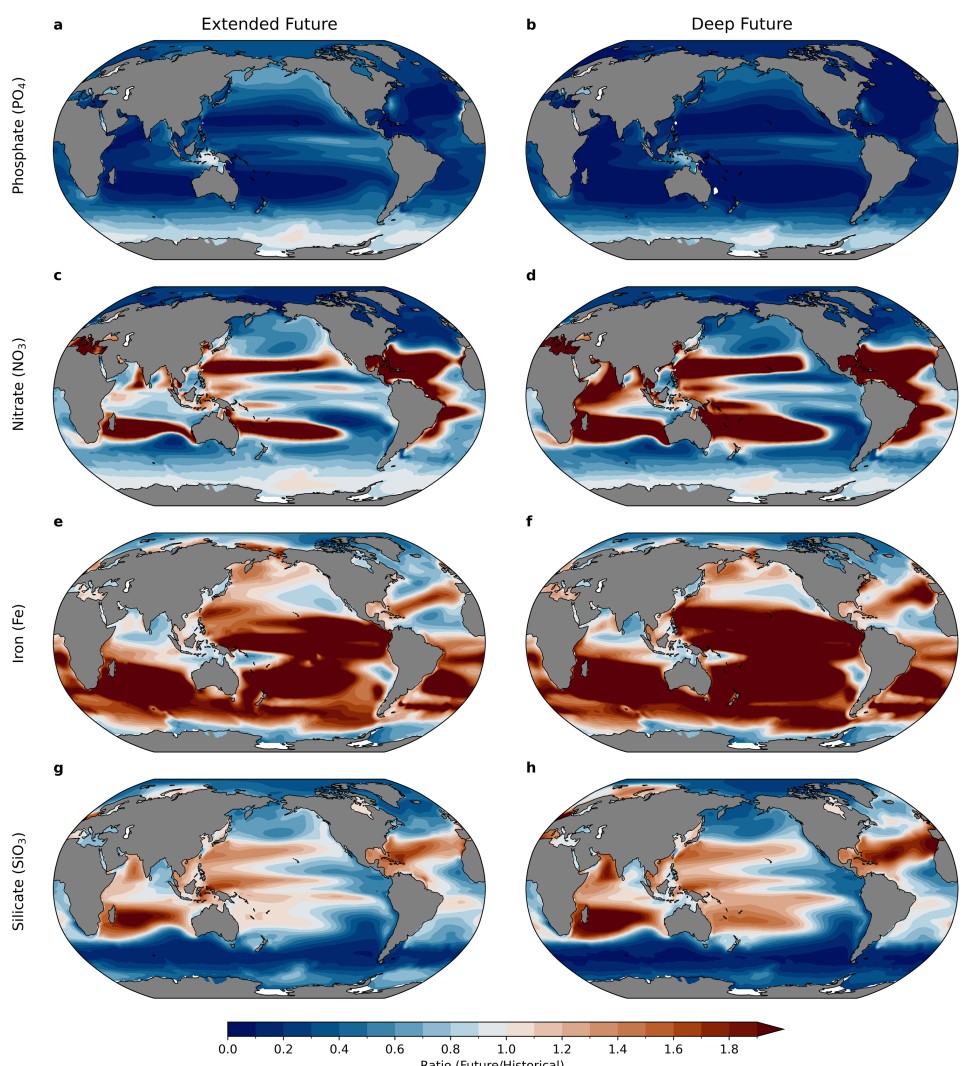

**Figure 10: The changes in the future period 2150-2249 (first column) and period 2401-2500 (second column) relative to the historical period 1850-1949. (a, b) PO$_4$ (mmol m$^{-3}$), (c, d) NO$_3$ (mmol m$^{-3}$), (e, f) Fe (mmol m$^{-3}$), and (g, h) SiO$_3$ (mmol m$^{-3}$). The changes were calculated as the ratio of future to historical periods. Blue (<1.0) and red (>1.0) indicate future decreases and increases, respectively, relative to the historical period. All values are averaged over the upper 100 m depth.**

We further examined the changes in the vertical structure of PO$_4$ and NO$_3$ (Fig. 11). The future changes indicate a substantial decrease in PO$_4$ concentrations, particularly above the thermocline, over the course of the simulations (Fig. 11b, c). The zonally averaged future changes in NO$_3$ concentrations show patches of increasing NO$_3$ concentrations in surface subtropical regions and decreases in the mesopelagic domain and higher latitudes (Fig. 11e, f). To understand the cause of the changes in zonally

averaged PO$_4$ shown in Fig. 11, we adopt the diagnostic approach applied by Rodgers et al. (2024). This approach considers the transfer efficiency of the mesopelagic domain by analyzing the ratio of fluxes at 1000 m to the fluxes at 100 m. Globally, the transfer efficiency for carbon over the recent historical period is approximately 13 % (Table 2). This value falls well within the uncertainty range provided by Doney et al. (2024), which they give as $12.2 \pm 4.1$ %. For both carbon and PO$_4$, the transfer efficiency changes only slightly by the end of the 25$^{th}$ century, stabilizing with a modest increase to a global value of

approximately 14 %. The modest simulated increases in transfer efficiency result from decreases in oxygen concentrations within the mesopelagic domain, which reduce the respiration rates of sinking organic particles. Meanwhile, the mesopelagic remineralization source, calculated as the difference between the fluxes at 100 m and 1000 m (rather than their ratio), shows substantial changes in the deep future, with decreases by the 2490s for carbon being 28.4 % and for PO$_4$ being 36.0 % (Table 2). The substantial decrease in PO$_4$ is largely due to the increased carbon-to-phosphorus ratios of sinking organic particles

under PO$_4$-depleted future conditions (Kwon et al., 2022), leading to disproportionately larger reductions in sinking particulate organic phosphorus than in sinking POC (Table 2). To understand how the PO$_4$ budget for the mesopelagic domain evolves,

it is important to consider that this largely reflects the cumulative (multi-century) impact of changes in remineralization rates, modulated by perturbations to boundary fluxes across the faces of the low-latitude mesopelagic domain. Although it is beyond the scope of the current study to fully close the budget, the closed budgets presented in Rodgers et al. (2024) for a steady-state case with a different state-of-the-art model suggest that the rates of material exchange of properties (mixing and advection) between the deep ocean and the mesopelagic domain within the tropics can also contribute to modulating low-latitude mesopelagic $PO_4$ inventories.

Notably, three key points should be emphasized regarding these changes in mesopelagic remineralization. First, the reduced rate of remineralization for $PO_4$ is consistent in sign with the projected decreases in zonally averaged $PO_4$ for the mesopelagic domain (Fig. 11b, c). Second, the future loss in $PO_4$ remineralization is greater than that for DIC. This characteristic of the plastic stoichiometric relationship between carbon and phosphate in MARBL indicates that the forced changes identified for $PO_4$ are partially buffered in the case of carbon. The partial buffering effect indicates the way in which the decrease in $PO_4$ remineralization does not fully translate into a proportional decrease in carbon remineralization. This implies that, while the remineralization of these two elements within the mesopelagic domain is related, it does not follow a fixed elemental ratio. In other words, the system buffers or dampens the expected changes in carbon fluxes that would have resulted from adherence to a fixed carbon-to-phosphorus ratio (Redfield et al., 1963), mitigating the overall impact of changes in phosphorus fluxes. The buffering or damping effect is approximately 20 % by the 2490s for the globally integrated impact (28.4 % for carbon and 36.0 % for $PO_4$). Third, despite warming of the mesopelagic domain for these simulations, the MARBL biogeochemistry does not respond with enhanced remineralization in response to increasing in situ temperatures. Consequently, this model does not provide a negative feedback through shoaling of the effective remineralization depth in response to mesopelagic warming that may compensate for decreases in export production in response to decreased overturning and increased stratification (Rodgers et al., 2024). Further analysis over the 30°S-30°N region shows that mesopelagic remineralization losses in this region are larger than the global mean, specifically 45.6 % for carbon and 53.8 % for $PO_4$ by the 2490s (Supplementary Table 1). In contrast, for the Southern Ocean domain spanning 90°S-60°S, mesopelagic remineralization sources for both carbon and $PO_4$ increase by 141 % by the 2490s (Supplementary Table 2). The results illustrate a contrast between these regions of reduced ocean overturning circulation, revealing opposite signs in their long-term responses. For the Southern Ocean, the response is thought to reflect changes in iron availability resulting from sea ice retreat (Moore et al., 2018). Interestingly, the modest buffering of the carbon mesopelagic remineralization source relative to $PO_4$ by plastic stoichiometry observed in low-latitude regions is not evident in the Southern Ocean region.

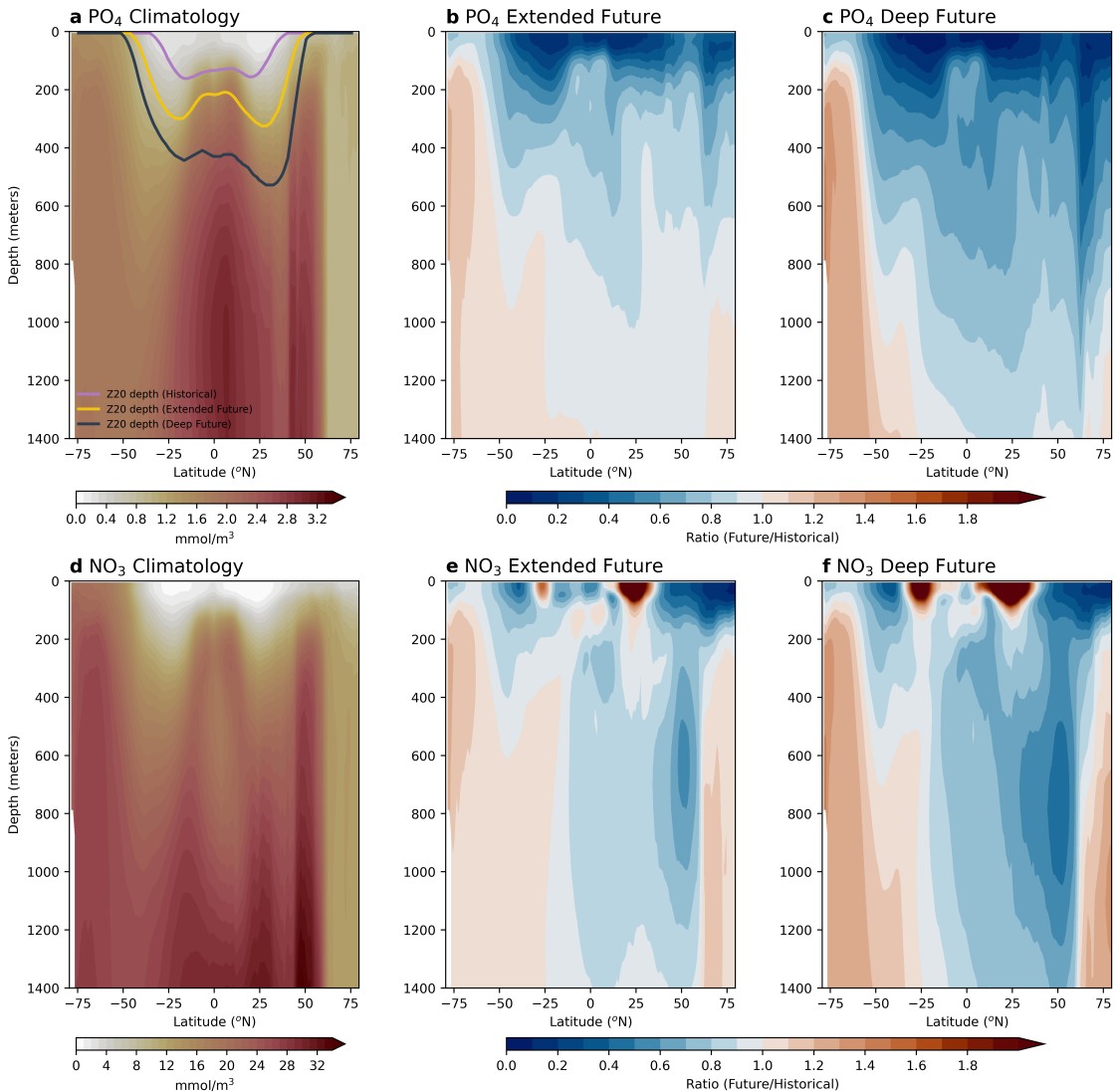

**Figure 11: Reference climatology (historical period 1850-1949) (first column), and changes in the future period 2150-2249 (second column) and period 2401-2500 (third column), relative to historical period 1850-1949. (a-c) Zonally averaged $PO_4$ (mmol m$^{-3}$) and (d-f) zonally averaged $NO_3$ (mmol m$^{-3}$). The changes in (b), (c), (e) and (f) were calculated as the ratio of future to historical periods. Blue (<1.0) and red (>1.0) in (b), (c), (e) and (f) indicate future decreases and increases, respectively, relative to the period 1850-1949. In (a), pink, yellow, and dark gray lines indicate the depth of 20°C isotherm averaged over the historical period (1850-1949), extended future period (2150-2249), and deep future period (2401-2500), respectively.**

**Table 2 The centennial-timescale ensemble-mean retention of mesopelagic (100 m-1000 m) carbon and $PO_4$ (italic and bold) over the global ocean domain. Fluxes of soft tissue at 100 m ($F_{100}$) and 1000 m ($F_{1000}$) represent fluxes at the top and bottom of the mesopelagic domain. Ten-year averages for each quantity are calculated over the 1890s, 1990s, 2090s, 2190s, 2290s, 2390s, and 2490s to minimize interannual variability beyond the 10-member ensemble average. Transfer efficiency is defined as the ratio of the fluxes at 1000 m and 100 m ($F_{1000}/F_{100}$), following Doney et al. (2024). The remineralization source is given as the difference between the fluxes at 100 m 1000 m ($F_{100} - F_{1000}$). For each case, the numbers given in parentheses are the percentage changes relative to the 1890s.**

| *GLOBAL*<br><br>DIC<br>***PO₄*** | $F_{100}$<br>(PgC yr$^{-1}$)<br>***(TmolPO₄ yr⁻¹)*** | $F_{1000}$<br>(PgC yr$^{-1}$)<br>***(TmolPO₄ yr⁻¹)*** | Transfer efficiency | Mesopelagic Accumulation<br>(Remineralization source)<br>(PgC yr$^{-1}$)<br>***(TmolPO₄ yr⁻¹)*** |
|---|---|---|---|---|
| 1890s | 6.98<br>***4.57*** | 0.936<br>***0.621*** | 0.134<br>***0.136*** | 6.04<br>***3.95*** |
| 1990s | 7.08 (+1.4%)<br>***4.61 (+0.8%)*** | 0.945 (+0.9%)<br>***0.627 (+0.9%)*** | 0.134<br>***0.136*** | 6.13 (+1.5%)<br>***3.98 (+0.7%)*** |

| | | | | |
|---|---|---|---|---|
| 2090s | 6.91 (-1.1%) *4.27 (-6.6%)* | 0.996 (+6.3%) *0.623 (+0.3%)* | 0.144 *0.146* | 5.91 (-2.2%) *3.65 (-7.7%)* |
| 2190s | 5.85 (-16.2%) *3.48 (-23.9%)* | 0.862 (-8.0%) *0.512 (-17.6%)* | 0.147 *0.147* | 4.99 (-17.5%) *2.97 (-24.8%)* |
| 2290s | 5.43 (-22.2%) *3.19 (-30.2%)* | 0.795 (-15.1%) *0.464 (-25.3%)* | 0.146 *0.145* | 4.63 (-23.3%) *2.73 (-31.0%)* |
| 2390s | 5.21 (-24.5%) *3.05 (-33.3%)* | 0.759 (-18.9%) *0.440 (-29.2%)* | 0.146 *0.144* | 4.45 (-26.4%) *2.61 (-34.0%)* |
| 2490s | 5.06 (-27.6%) *2.95 (-35.5%)* | 0.732 (-21.8%) *0.422 (-32.0%)* | 0.145 *0.143* | 4.33 (-28.4%) *2.53 (-36.0%)* |

Finally, diagnostics were performed of the export ratio for biogeochemical fluxes, defined as the ratio of the sinking or export production flux to NPP (Supplementary Table 3). Over the global domain, the ensemble-mean value for the 1990s in our simulations is 0.145. This falls within the uncertainty bounds of the levels presented by Doney et al. (2024), which are 0.154 ± 0.026 for the forced ocean biogeochemical models included in the REgional Carbon Cycle Assessment and Processes

(RECCAP2) project. This ratio over the global domain is projected to decrease by 16.6 % by the 2490s, indicating that nutrient recycling in the euphotic zone is enhanced over time. In the 30°S-30°N domain, the decrease of 24.0 % by the 2490s is larger than the global mean reduction. This result is consistent in sign with the finding from earlier CESM1 simulations under similarly strong CMIP5 forcing (Moore et al., 2018), and can be partly attributed to a larger fraction of NPP being allocated to dissolved organic matter rather than particulate organic matter under a warmer climate (Sreeush et al., 2024). It is worth

noting that the extratropics of the Southern Hemisphere (90°S-30°S) and Northern Hemisphere (30°N-90°N) show quite different trends. The export ratio decreases by only 4.8 % in the Southern Hemisphere extratropics, while it decreases by 28.1 % in the Northern Hemisphere extratropics.

We further examined the evolution and the spatial patterns of ocean carbon uptake in response to greenhouse warming. The global integral of ocean carbon uptake shows a peak around 2100 in excess of 4 PgC yr$^{-1}$, followed by an exponential decline

to less than 1 PgC yr$^{-1}$ by 2500 (Fig. 2f, Fig. 12a). Our simulations show that, until the 23$^{rd}$ century, the uptake over the latitude range of the subtropical cells (45°S-40°N) contributes at least half of the total uptake, however, by 2500, this contribution diminishes to less than 25 %. The Southern Ocean (90°S-45°S) accounts for about 25 % of the total uptake at the end of the 21$^{st}$ century, then this contribution increases to nearly half of the global uptake by 2500. The northern latitudes (40°N-90°N) show peak uptake during the 21$^{st}$ century, but then decline to less than 0.5 PgC yr$^{-1}$ after 2200. To better understand the changes

in ocean carbon uptake, we investigated the D$p$CO$_2$ distribution (the difference in $p$CO$_2$ between the ocean and atmosphere). During the historical reference period 1850-1949, the thermodynamic gradient in $p$CO$_2$ driving CO$_2$ outgassing was largest for the equatorial and eastern boundary upwelling regions of the Pacific and Indian Oceans (Fig. 12b), whereas for the deep future period this distribution shifts, extending further into the Indian Ocean and showing maxima near 20° latitude in both hemispheres (Fig. 12c). To shed light on how changes in the ocean state impact the natural carbon cycle, we examined the

CO$_2$ fluxes for RAD (radiatively coupled forcing). Instead of a gradual change towards warming-induced CO$_2$ outgassing, the global integral shows a shift towards net outgassing of natural carbon, which occurs predominantly between 2100 and 2200 (Fig. 12d). This decline is largely driven by a longer-term trend towards net outgassing in the subtropical regions, with a smaller contribution that occurs earlier in the northern high latitudes. In contrast, the Southern Ocean shows a shift towards enhanced CO$_2$ uptake after the 22$^{nd}$ century. Our further analysis of RAD reveals a shift towards broad CO$_2$ outgassing over

the region spanning 30°S-30°N (Fig. 12f), in accordance with large reductions in POC export over those areas (Fig. 9f). The enhanced CO$_2$ uptake over the Southern Ocean is also partly driven by the future strengthening of biological carbon pump, which results in sequestering more DIC in the deep ocean. Since this natural carbon tracer only sees pre-industrial atmospheric

$CO_2$ levels through the gas exchange by construction, this shift to outgassing is indicative of a positive climate feedback by the marine carbon cycle, operating through the natural carbon cycle (Koeve et al., 2024), that could counteract the overall carbon uptake shown in Fig. 12a.

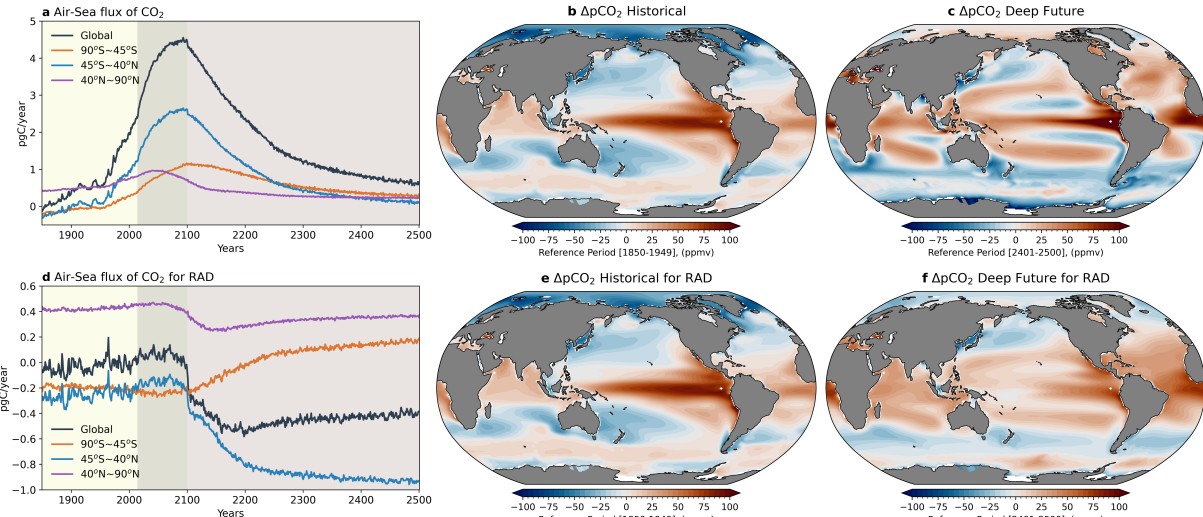

**Figure 12: Changes in surface ocean carbon state and fluxes. (a) Spatially integrated air-sea $CO_2$ fluxes (PgC year$^{-1}$) for the ensemble mean considered globally (black), 90°S-45°S (orange), over 45°S-40°N (blue), and over 40°N-90°N (purple), where a 12-month running mean has been applied to the ensemble mean to remove seasonal variations, and positive values indicating ocean uptake of carbon. (b) $\Delta p CO_2$ (ppmv) averaged over the 1850-1949, taken as the difference between oceanic and atmospheric $p CO_2$ and (c) $\Delta p CO_2$ averaged over the 2401-2500. Positive values occur where there is outgassing. (d-f) Same as (a-c) except for RAD case (i.e., the system is radiatively coupled, but pre-industrial $CO_2$ concentrations are maintained for air-sea gas exchange).**

## 4. Summary and Discussion

Sustained anthropogenic warming has widespread and profound impacts on the Earth's climate system, and understanding these effects beyond 2100 is crucial for highlighting potential future natural disaster risks and assessing long-term climate sensitivity. To explore multi-centennial responses to sustained warming over the next few centuries, we have conducted new long-term ensemble simulations which extend 10 members of CESM2-LE to the year 2500. Our fully coupled simulations, among the most comprehensive in scope for future projections, provide insights into changes in the mean state and variability for key fields across the atmosphere, ocean, sea ice, land, and ecosystems on multi-centennial time scales from 1850 to 2500. Additionally, we discuss potential carbon-climate feedbacks that may emerge beyond 2100.

Over the period 1850-2500, our extended simulations project a strong greenhouse warming of greater than 12 °C by the end of the 25[th] century relative to the historical period, despite the fact that fossil and industrial $CO_2$ emissions are effectively reduced to zero by 2250 under the extended SSP3-7.0 scenario. By the 25[th] century, the eastern equatorial Pacific is projected to warm by 8-10 °C relative to the historical reference period 1850-1949. This warming will be accompanied by weakened easterlies, intensified atmospheric upward motion over the entire equatorial Pacific sector, and increased precipitation within a narrower latitudinal range characterized by a northward shift of the SPCZ with a loss of its diagonal orientation toward the subtropics (Fig. 3). These changes in the mean state could substantially alter ocean-atmosphere interactions, including the Pacific Walker circulation. In addition to the mean precipitation changes, at a local scale, extreme precipitation events may become more frequent across different geographical regions, while diverse changes in the timing and intensity of the seasonal distribution of precipitation are projected to occur (Fig. 6), indicating heterogeneous impacts of climate change at regional and urban levels.

Amplified warming in the Northern Hemisphere high latitudes will drive permafrost thawing, which is a major pathway for loss of soil carbon after the mid-22nd century, and thus the land will subsequently become a source of $CO_2$. However, on multi-centennial timescales, the global ocean still acts as a sink for anthropogenic carbon, with the increases of Southern Ocean's contribution from 25 % to nearly 50 % by 2500 (Fig. 12), although continuous reduction in ocean carbon uptake is projected from the 22nd century due to decreases in the $CO_2$ buffering capacity of seawater. Our CESM2-LE extension simulations under SSP3-7.0, consistent with the findings of Rodgers et al. (2024) for projections with CESM2-WACCM to 2299 under SSP5-8.5 forcing, project a substantial depletion of $PO_4$ in the low-latitude mesopelagic domain, along with a modest increase in the transfer efficiency. This decrease in $PO_4$ remineralization leads to substantial surface $PO_4$ depletion and reduced NPP in subtropical gyres from the 22nd century onward. The projected future changes in nutrient dynamics could affect primary productivity, food webs, and biodiversity in marine ecosystems.

Through analyses focused on how sustained warming can impact variance within the Earth system, we identified decreases in surface temperature variability across most regions except for Africa, Asia, Australia, South America, and the North Atlantic. In contrast, precipitation variability is projected to increase in most areas, particularly in the tropics where the merging of precipitation bands amplifies variability. The model projects substantial reductions in ENSO variability beyond 2100, but precipitation variability in the Niño3.4 region shows a rebound after the mid-23rd century. In contrast to ENSO variability changes, the intensity of the MJO in our extended simulations is projected to strengthen notably, making it a key player in representing variability in the tropical region during the post-2100 period (Fig. 5).

The Earth system is currently on a Hothouse Earth trajectory driven by human-induced greenhouse gas emissions (Steffen et al., 2018). Our simulations project climate states that are well outside the range of what humans have thus far experienced, making it challenging to present the astonishingly broad range of large impacts under sustained anthropogenic perturbations. We emphasize that our study aims to provide a comprehensive overview of these important changes. Given the considerable uncertainty in future decisions regarding anthropogenic emissions, we do not intend to suggest that the climate changes projected in our simulations are likely pathways for the evolution of climate states. However, our simulations may serve as a crucial illustration of a potential future that we should strive to avoid. A novel aspect of our ensemble simulations lies in the presentation of not only mean state changes but also of shifts in the variability of the Earth system across diverse temporal and spatial scales under the influence of strong anthropogenic forcing. Furthermore, we anticipate that the analyses presented here can contribute a foundational step toward understanding the mechanisms that can impact multi-century climate-carbon feedbacks. Further complementary simulations, as well as emission-driven scenarios (Sanderson et al., 2024), are needed to explore these potential feedbacks more thoroughly. Although the future climate projections beyond 2100 are influenced by multiple factors such as the climate sensitivity of models and the choice of emission scenarios, it is apparent that there is an urgent need for drastic reductions in carbon emissions from human activities. Our study indicates that developing long-term mitigation policies based on the post-2100 perspective is not mature.

**Code availability**

The CESM2 code is available from https://www.cesm.ucar.edu/models/cesm2. Scripts and data to reproduce the figures and analysis of this paper can be found at https://climatedata.ibs.re.kr/data/papers/lee-et-al-2024-earth-system-dynamics (will be updated after review).

**Data availability**

All datasets used in the study are publicly available. CESM2-LE extension simulations datasets are available on ICCP Climate Data Website (will be in https://ibsclimate.org after review).

## 670    Author contributions

The model simulations were set up and performed through a collaborative effort by S.-S.L. and N.R. The scientific framework of this manuscript was developed by S.-S.L., S.S, and K.B.R.. All authors discussed the results and contributed to the analysis and writing of the manuscript.

## 675    Competing interests

At least one of the (co-)authors is a member of the editorial board of Earth System Dynamics.

## Acknowledgements

We thank A. Timmermann, Gokhan Danabasoglu, Keith Lindsay, and Michael Mills for their valuable comments, discussions
and support throughout this work. This work was supported by the Institute for Basic Science (IBS), Republic of Korea, under IBS-R028-D1. The simulations presented here were carried out on the IBS/ICCP supercomputer "Aleph", a 1.43 petaflop high-performance Cray XC50-LC Skylake computing system with 18,720 processor cores, 9.59 PB of disc storage, and 43 PB of tape archive storage. We also acknowledge the support of KREONET.

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
