# Peer review of "Multi-centennial climate change in a warming world beyond 2100"

_EGUsphere, 2024_

## Author Comment (AC1)

**Response to Reviewer #1**

GENERAL COMMENTS

This paper describes the changes in the physical climate system as well as the biochemical responses in climate projections with CESM2 extended to 2500 under SSP3-7.0. In this extended scenario, $CO_2$ emissions increase until 2100, then follow a path of $CO_2$ emission reductions until 2250 when net-zero emissions of $CO_2$ are reached. The simulations reach a warming approaching 12°C by ~2300 and stabilise at that temperature by 2500.

The paper is very well written and nicely laid out, and the results are interesting and a nice addition to the literature on multi-century climate projections. It is in some places quite dense to get through, but overall, I recommend publication after minor revisions. The geochemistry sections in particular are a bit difficult to understand in some places for an interdisciplinary / non-specialist readership.

One point that I would like to see briefly commented on, is some context on the relevance of this scenario for future projections and its implications. In e.g. the section on rainfall seasonality changes in megacities, L299 states "potentially resulting in substantial social and economic losses". While this is of course true, I think this is a vast understatement of the situation/problem at +12C and we can no longer talk about "potential losses and impacts" in such a hugely different simulated future. Without looking up an exact definition, and while this is of course subjective and not an exact science, I think people usually consider projected changes >3-4°C as "catastrophic" (if not sooner). 12°C is far beyond that and would likely include fundamental changes to the climate system. How is the reader meant to interpret those projections? A bit of context here would be welcome.

A: We would like to thank the reviewer for their thoughtful evaluation of our manuscript, as well as for their constructive comments and suggestions. We have carefully revised the manuscript to incorporate the suggested improvements. Additionally, we have made every effort to present the content in a clear and accessible manner for readers. As the reviewer notes, most individuals would consider a warming of over 3-4°C to be catastrophic. Our simulations reflect a climate state that has never been experienced by humans, making it challenging to discuss specific impacts and losses. However, we believe our simulations still serve as an important illustration of a potential future that we should strive to avoid. The promotion of assessments of the Earth system evolution under such changes over these timescales is also likely to become one of the CMIP7 protocols, in recognition of the high value of such assessments for informing policy and awareness of the risks and consequences of weak mitigation. Therefore, exploring the potential climate changes and impacts of such a scenario is both useful and necessary as a warning. We also included this point in the revised manuscript.

Our detailed point-by-point replies (in black) to reviewer's comments (in blue) are given below.

SPECIFIC COMMENTS

I include specific comments and requests for clarifications below.

1. L227 "the weakening of the AMOC is related to the southward shift of the ITCZ (references)": the sentence here reads as if the ITCZ shift causes the AMOC weakening, but surely that is not what you mean? Is it the other way around? Do you have any evidence for this other than the references? Please rephrase this sentence to include more details, and perhaps something like "studies have linked this change to …" to indicate this is not something that you have shown here.

A: We thank the reviewer for this comment. Following the reviewer's suggestion, we have rephrased the text in the revised manuscript:

"Additionally, our simulations project a southward shift of the climatological Intertropical Convergence Zone (ITCZ), as part of the coupled climate system response to the AMOC slowdown as indicated by previous studies (e.g., Bellomo et al., 2021; Vellinga and Wood, 2008; Zhang and Delworth, 2005), resulting in substantial precipitation increase along the equator and merging of two tropical precipitation bands. As a result, tropical precipitation zones are projected to become narrower (Lau and Kim, 2015) and the dry regions between the two tropical convergence zones are likely to experience wetter conditions in the post-2100 (Fig. S2a-c, Fig. 3d-f)"

2. L235-236 "In addition to the weakening of the easterlies […], our simulations project a substantial reduction of the trade winds in the extended future […]": isn't a weakening in the easterlies the same as a reduction in the trade winds, so you are repeating the same thing twice? Or is there a difference?

A. We thank the reviewer for pointing this out. We have rephrased the text to clarify the description as follows:

"Our simulations project an overall weakening of the easterlies in the Pacific and Atlantic, spanning approximately 30°S to 30°N, with a substantial reduction along the equator in the Pacific in the extended future. This also reflects the weakening of Walker circulation. These changes are projected to become even more pronounced in the deep future (Fig. 3k, l)."

3. L257, "disappearance of the SPCZ": I think this statement is too strong without further evidence, but the Supplementary Material supports this statement. Can you include a reference to the SI? Figure 3e and f) alone are not enough, as they show the difference relative to the reference climatology, not a new climatology for the later periods.

A: We thank the reviewer for this comment. We have added a reference to the relevant text as well as for Fig. S2b and c regarding future changes in the SPCZ. Additionally, given the issue of model-dependency of SPCZ changes in future projections (Narsey et al., 2022), we have modified the text in the revised manuscript as follows and have made efforts to avoid making overly strong or forceful statements without sufficient evidence:

"One notable feature of the tropical precipitation changes in our extended simulations is the retreat or disappearance of the climatological South Pacific Convergence Zone (SPCZ), which is characterized by a northward shift with loss of its diagonal orientation towards the subtropics relative to the historical climatology (Fig. S2b, c). A comparison between 50-year periods (1950-99 and 2050-99) reveals that several CMIP5 and CMIP6 models show a northward shift of the SPCZ in the future, consistent with our model projections, while others predict a southward shift (Narsey et al., 2022). The equatorward shift and the projected changes in the zonal structure of the SPCZ can be linked to the SST response in the central equatorial Pacific (Cai et al., 2012). Additionally, our simulations project a southward shift of the climatological Intertropical Convergence Zone (ITCZ), as part of the coupled climate system response to the AMOC slowdown as indicated by previous studies (e.g., Bellomo et al., 2021; Vellinga and Wood, 2008; Zhang and Delworth, 2005), resulting in substantial precipitation increase along the equator and merging of two tropical precipitation bands. As a result, tropical precipitation zones are projected to become narrower (Lau and Kim, 2015) and the dry regions between the two tropical convergence zones are likely to experience wetter conditions in the post-2100 (Fig. S2a-c, Fig. 3d-f)."

4. Figure 4h: Title should be precipitation instead of Nino3.4 SST variability (it is the same title as panel d).

A: We have modified the title in the manuscript.

5. E.g. Figure 10: "Fractional changes" typically refer to changes relative to a reference period, e.g. (Extended future – historical)/historical, rather than just a ratio between two periods. Perhaps rephrase to "ratio between Period2/Period1? Also, fractional changes are normally unitless.

A: We thank the reviewer for pointing this out. We have modified the text and the figure caption accordingly in the revised manuscript, as shown here:

[Figure]

**Figure 10: The changes in the future period 2150-2249 (first column) and period 2401-2500 (second column) relative to the**

**historical period 1850-1949. (a, b) $PO_4$ (mmol m$^{-3}$), (c, d) $NO_3$ (mmol m$^{-3}$), (e, f) Fe (mmol m$^{-3}$), and (g, h) $SiO_3$ (mmol m$^{-3}$).**

**The changes were calculated as the ratio of future to historical period. Blue (<1.0) and red (>1.0) indicate future decreases**

**and increases, respectively, relative to the historical period. All values are averaged over the upper 100 m depth.**

6. L339: this should be "availability of food" or "food security", not both.

A: We thank the reviewer for pointing this out, and we have modified the manuscript accordingly ('security' has been deleted).

7. L389-390: I was wondering up to this point whether the simulations were concentration or emissions-driven. Please also include this information in the section describing the simulations.

A: Our simulations are concentration-driven simulations, thus the greenhouse gas concentrations (defined as dry air mole fractions) from historical observations and the SSP3-7.0 scenario were prescribed over the entire simulation period (e.g., regarding $CO_2$, atmospheric $CO_2$ mole fraction is specified directly from 1850 to 2500). We have clarified this point in Section 2:

"Meinshausen et al. (2020) provided greenhouse gas concentrations, defined as dry air mole fractions, for both standard and extended SSP scenarios. They used the reduced-complexity climate-carbon-cycle model MAGICC7.0 ('Model for the Assessment of Greenhouse Gas Induced Climate Change') to produce future greenhouse gas concentrations driven by harmonized SSP greenhouse gas emissions (Gidden et al., 2019) and extended emissions beyond 2100. To extend the CESM2-LE from 2101 to 2500, we followed the extended SSP3-7.0 protocol, a concentration-driven configuration. In this extended scenario, fossil and industrial $CO_2$ emissions are effectively ramped down to zero by 2250 (Meinshausen et al., 2020), as shown in Fig. 1a. Figure 1b presents the time evolution of global mean greenhouse gas mole fractions ($CO_2$, $CH_4$, $N_2O$, and CFCs) which are prescribed in these simulations under the historical (1850-2014), standard SSP3-7.0 (2015-2100) and extended SSP3-7.0 (2101-2500) scenario forcings. The global mean atmospheric $CO_2$ mole fraction at the end of the 25[th] century provided by the extended SSP3-7.0 scenario is approximately 1371 ppm."

8. L414: Please be more explicit about the link between NPP and POC export for non-biochemistry specialists. Does export imply a direction (vertical downwards)?

A: We thank the reviewer for pointing out the need to improve the clarity of our explanation in order to be accessible to non-specialists.

"Next, we revisit the approximately 15 % drop in globally integrated marine NPP shown in Fig.

2e. As a substantial fraction of newly produced organic matter is vertically exported below the surface and remineralized into inorganic matter in the ocean's interior - a process known as the biological pump - NPP changes within the euphotic zone propagate into changes in particulate organic carbon (POC) fluxes throughout the water column. In other words, export represents a net downward transport of organic material (typically expressed in carbon mass units) across a depth horizon, with this component of the flux being primarily sustained through gravitational sinking.

Primary production is the rate at which organic material is produced from inorganic compounds and does not have a directional component. Given the predominance of recycling within the sunlit surface layers of the ocean, primary production is almost always larger than export. Regionally, ocean NPP increases by up to ~80 % in the Southern Ocean and decreases by up to 50 % in the mid-latitudes of the North Atlantic in the extended future (Fig. 9b). The spatial patterns of NPP

change are qualitatively similar to the changes in POC export at a depth of 100 m (Fig. 9e), ….."

9. L453-454 "thermocline $PO_4$ concentrations": do you mean concentrations above the thermocline? Suggest adding a line highlighting the position of the thermocline to Figure 11.

A: Yes, the text refers to the $PO_4$ concentrations above the thermocline. We have clarified this in the revised manuscript. In addition, following the reviewer's suggestion, we have added lines representing the depth of the 20°C isotherm averaged over the historical period (1850-1949), the extended future period (2150-2249), and the deep future period (2401-2500) to Fig. 11a in the revised manuscript to represent the position of the thermocline, as shown here:

[Figure]

**Figure 11: Reference climatology (period 1850-1949) (first column), and changes in the period 2150-2249 (second column) and period 2401-2500 (third column), relative to 1850-1949. (a-c) Zonally averaged PO$_4$ (mmol m$^{-3}$) and (d-f) zonally averaged NO$_3$ (mmol m$^{-3}$). The changes in b, c, e and f were calculated by dividing the values in the future periods by the values in the period 1850-1949. Blue (<1.0) and red (>1.0) in b, c, e and f indicate future decreases and increases, respectively, relative to the period 1850-1949. In a, pink, yellow, and dark gray lines indicate the depth of 20°C isotherm averaged over the historical period (1850-1949), extended future period (2150-2249), and deep future period (2401-2500), respectively.**

10. L462 "Viewed this way": what way? This is unclear – the previous sentence states that the change in the transfer efficiency is modest, yet here you state the changes in the remineralisation source are substantial. Please clarify.

A: We apologize for any confusion. We have added an additional short paragraph:

"For both carbon and PO$_4$, the transfer efficiency changes only slightly by the end of the 25$^{th}$

century, stabilizing with a modest increase to a global value of approximately 14 %. The modest
simulated increases in transfer efficiency result from decreases in oxygen concentrations within
the mesopelagic domain, which reduce the respiration rates of sinking organic particles.
Meanwhile, the mesopelagic remineralization source, calculated as the difference between the
fluxes at 100 m and 1000 m (rather than their ratio), shows substantial changes in the deep future.
By the 2490s, decreases are projected to be 28.4 % for carbon and 36.0 % for $PO_4$ (Table 2). The
substantial decrease in $PO_4$ is largely due to the increased carbon-to-phosphorus ratios of sinking
organic particles under $PO_4$-depleted future conditions (Kwon et al., 2022), leading to
disproportionately larger reductions in sinking particulate organic phosphorus than in sinking
particulate organic carbon (Table 2). To understand how the $PO_4$ budget for the mesopelagic
domain evolves, it is important to consider that this largely reflects the cumulative (multi-century)
impact of changes in remineralization rates, modulated by perturbations to boundary fluxes across
the faces of the low-latitude mesopelagic domain. Although it is beyond the scope of the current
study to fully close the budget, the closed budgets presented in Rodgers et al. (2024) for a steady-
state case with a different state-of-the-art model suggest that the rates of material exchange of
properties (mixing and advection) between the deep ocean and the mesopelagic domain within the
tropics can also contribute to modulating low-latitude mesopelagic $PO_4$ inventories."
11. L468: Are there any explanations for this partial buffering in the case of carbon?
A: In response to the reviewer's comment, we have added an explanation for the partial buffering
effect in the revised manuscript:
"Second, the fractional loss in $PO_4$ remineralization is greater than that for DIC. This characteristic
of the plastic stoichiometric relationship between carbon and phosphate in MARBL indicates that
the forced changes identified for $PO_4$ are partially buffered in the case of carbon. The partial
buffering effect indicates the way in which the decrease in $PO_4$ remineralization does not fully
translate into a proportional decrease in carbon remineralization. This suggests that while the
remineralization of these two elements within the mesopelagic domain is related, it does not follow
with a fixed elemental ratio. In other words, the system buffers or dampens the expected changes
in carbon fluxes that would have resulted from adherence to a fixed carbon-to-phosphorus ratio
(Redfield et al., 1963), mitigating the overall impact of changes in phosphorus fluxes. Third, despite warming of the mesopelagic domain for these simulations, ….."

12. L548: "unprecedented": there are a handful of studies looking at very long future projections at different global warming levels with fully coupled models, though the experimental design is different. Not a criticism, just mentioning for awareness.

A: We thank the reviewer for their comment. We have rephrased this sentence as follows:

"Our fully coupled simulations, among the most comprehensive in scope for future projections, provide insights into changes in the mean state and variability for key fields across the atmosphere, ocean, sea ice, land, and ecosystems on multi-centennial time scales from 1850 to 2500."

13. L572-573 ". the stoichiometric plasticity mechanism identified by Kwon et al. (2022) provides only moderate modulation into the deep future." What is the mechanism identified by Kwon et al.?

Moderate modulation of what?

A: We have included a detailed explanation in Section 3.3 and made revisions to the text in Section

4 of the revised manuscript:

- Line 422 in the submitted version:

"The NPP reductions in oligotrophic gyres are also driven by surface nutrient depletion, suggesting that nutrient uptake plasticity – an adaptive strategy used by phytoplankton to reduce nutrient uptake while maintaining carbon fixation (i.e., increasing carbon-to-phosphorus ratios in phytoplankton cells or communities) when surface nutrients are scarce - is less effective beyond the 21st century. This strategy, which plays a key role in sustaining ocean NPP until the 21st century (Kwon et al., 2022), loses its effectiveness in later periods."

- Line 571 in the submitted version:

"This decrease in $PO_4$ remineralization leads to substantial surface $PO_4$ depletion and reduced NPP

in subtropical gyres from the 22nd century onward. The projected future changes in nutrient dynamics could affect primary productivity, food webs, and biodiversity in marine ecosystems."

**References**

Bellomo, K., Angeloni, M., Corti, S., and von Hardenberg, J.: Future climate change shaped by inter-model differences in Atlantic meridional overturning circulation response, Nature Communications, 12, 3659, 10.1038/s41467-021-24015-w, 2021.

Cai, W., Lengaigne, M., Borlace, S., Collins, M., Cowan, T., McPhaden, M. J., Timmermann, A., Power, S., Brown, J., and Menkes, C.: More extreme swings of the South Pacific convergence zone due to greenhouse warming, Nature, 488, 365-369, 2012.

Gidden, M. J., Riahi, K., Smith, S. J., Fujimori, S., Luderer, G., Kriegler, E., van Vuuren, D. P., van den Berg, M., Feng, L., Klein, D., Calvin, K., Doelman, J. C., Frank, S., Fricko, O., Harmsen, M., Hasegawa, T., Havlik, P., Hilaire, J., Hoesly, R., Horing, J., Popp, A., Stehfest, E., and Takahashi, K.: Global emissions pathways under different socioeconomic scenarios for use in CMIP6: a dataset of harmonized emissions trajectories through the end of the century, Geosci. Model Dev., 12, 1443-1475, 10.5194/gmd-12-1443-2019, 2019.

Kwon, E. Y., Sreeush, M. G., Timmermann, A., Karl, D. M., Church, M. J., Lee, S.-S., and Yamaguchi, R.: Nutrient uptake plasticity in phytoplankton sustains future ocean net primary production, Science Advances, 8, eadd2475, doi:10.1126/sciadv.add2475, 2022.

Lau, W. K. M. and Kim, K.-M.: Robust Hadley Circulation changes and increasing global dryness due to $CO_2$ warming from CMIP5 model projections, Proceedings of the National Academy of Sciences, 112, 3630-3635, doi:10.1073/pnas.1418682112, 2015.

Meinshausen, M., Nicholls, Z. R. J., Lewis, J., Gidden, M. J., Vogel, E., Freund, M., Beyerle, U., Gessner, C., Nauels, A., Bauer, N., Canadell, J. G., Daniel, J. S., John, A., Krummel, P. B., Luderer, G., Meinshausen, N., Montzka, S. A., Rayner, P. J., Reimann, S., Smith, S. J., van den Berg, M., Velders, G. J. M., Vollmer, M. K., and Wang, R. H. J.: The shared socio-economic pathway (SSP) greenhouse gas concentrations and their extensions to 2500, Geosci. Model Dev., 13, 3571-3605, 10.5194/gmd-13-3571-2020, 2020.

Narsey, S., Brown, J. R., Delage, F., Boschat, G., Grose, M., Colman, R., and Power, S.: Storylines of South Pacific Convergence Zone Changes in a Warmer World, Journal of Climate, 35, 6549-6567, https://doi.org/10.1175/JCLI-D-21-0433.1, 2022.

Redfield, A. C., Ketchum, B. H., and Richards, F. A.: The influence of organisms on the composition of seawater, The sea, 2, 26-77, 1963.

Rodgers, K. B., Aumont, O., Toyama, K., Resplandy, L., Ishii, M., Nakano, T., Sasano, D., Bianchi, D., and Yamaguchi, R.: Low-latitude mesopelagic nutrient recycling controls productivity and export, Nature, 632, 802-807, 10.1038/s41586-024-07779-1, 2024.

Vellinga, M. and Wood, R. A.: Impacts of thermohaline circulation shutdown in the twenty-first century, Climatic Change, 91, 43-63, 2008.

Zhang, R. and Delworth, T. L.: Simulated Tropical Response to a Substantial Weakening of the Atlantic Thermohaline Circulation, Journal of Climate, 18, 1853-1860, https://doi.org/10.1175/JCLI3460.1, 2005.

---

## Author Comment (AC2)

**Response to Reviewer #2**

GENERAL COMMENTS

In this study, a set of extended simulations under forcings equivalent to reduced and then net zero carbon emissions is generated for the CESM2 model and then these are analysed. Substantial long-term climate change signals are identified in terms of temperature and precipitation, climate variability, carbon fluxes, and ocean nutrients.

The study is comprehensive and a useful addition to the literature. I don't have any major concerns, although I have quite a few comments and suggestions below. Most of the comments are around the framing of the analysis and some overly strong inferences in my opinion.

I did think that even though a selling point of the analysis is the ensemble of simulations, much of the analysis doesn't really make the most of the large sample sizes available. This is a minor comment though and I would be keen to see follow up analyses where these simulations are used to look at climate extremes for example.

A: We sincerely thank the reviewer for evaluating our manuscript and providing a valuable number of comments and suggestions. In this study, we mainly focused on the forced changes (i.e., 10-ensemble mean) resulting from anthropogenic forcings with somewhat less emphasis on the ensemble spread and probability distribution (e.g., Fig. 2, Fig. 4, and Fig. 6). Nevertheless, we wish to emphasize that we did include analyses and interpretations on forced changes in the variability of the system, including not only ENSO timescales but also seasonal cycles of precipitation and marine sea surface $pCO_2$, as well as with the MJO.

To further advance our understanding of the probability distribution of the long-term mean state changes under greenhouse warming, additional analysis will be conducted and presented in future studies. Given the priorities of a scientific overview with the current study, we opted to emphasize the scientific novelty of forced changes in variability, where we have leveraged the wealth of ensemble information in the simulations. In fact, a more comprehensive analysis of climate extremes and attribution of their mean-state drivers under sustained warming is underway as a complementary publication. Additionally, we are conducting parallel research on forced changes in monsoonal behavior over multi-centennial timescales, where the probability information from the ensembles will be more exhaustively considered.

We have carefully revised the manuscript to incorporate the reviewer's comments and suggestions. Below, we present our detailed point-by-point responses (in black) to the reviewer's comments (in blue).

SPECIFIC COMMENTS

1. Line 11: The first sentence could be clearer. Maybe just "Changes in the climate system are anticipated well beyond the 21$^{st}$ century due to human influences." Or something similar?

A: We thank the reviewer for their comments here. We have modified the sentence as follows:

"Changes in the climate due to human influences are expected to extend well beyond the 21$^{st}$ century."

2. L18-19: I don't think this sentence is very helpful without more clarification on the baseline and given likely influence of high climate sensitivity on these numbers. I'd suggest a more qualitative sentence about global mean temperature and precipitation remaining elevated under net zero emissions would be more useful.

A: Following the reviewer's suggestion, we have modified the text:

"Global mean surface temperature and precipitation are projected to continue rising even after $CO_2$ emissions cease."

3. L23: Please define $PO_4$.

A: $PO_4$ refers to dissolved inorganic phosphate. We have revised the text to explicitly clarify this.

4. L24-26: This is a very confusing sentence that appears to try and summarise quite different findings in one. Some editing of this would be helpful.

A: We have rephrased this sentence as follows:

"The extended simulations predict substantial changes in the amplitude and timing of precipitation seasonality at the urban scale, with variations across different locations. Similarly, seasonal variations in the partial pressure of $CO_2$ in seawater along different latitudinal bands are projected to experience distinct changes. These findings suggest that post-2100 changes will not simply be an extension of the trends projected for the 21$^{st}$ century.

5. L29-30: You might want to note CMIP7 plans (van Vuuren et al., 2025) which have standard scenario runs out to 2125 and more emphasis on extensions partly in response to the issue you raise.

A: We thank the reviewer for the comment. We have added this study to the revised manuscript:

"…… protocols have been developed to extend Coupled Model Intercomparison Project (CMIP)

simulations to 2500 Meinshausen et al. (2020). Very recently, scenarios for long-term extensions up to 2500 have been newly proposed for the ScenarioMIP experiments of CMIP phase 7

(ScenarioMIP-CMIP7) (van Vuuren et al., 2025). The growing interest in these extended timescales can be seen….."

6. L45-54: Santana-Falcón et al., (2023) may also be a relevant paper to use for this point, albeit with stronger mitigation than studied in this paper.

A: We thank the reviewer for bringing this study to our attention. We have included it in the relevant text of the revised manuscript (line 148 in the submitted version):

"By the end of the 25th century, the projections indicate a cumulative increase of ~17,000 ZJ, which is approximately seven times higher than the projected heat content perturbation by the end of the 21st century in the CESM2-LE (Rodgers et al., 2021). Based on the idealized and comprehensive overshoot simulations, it is noted that human-induced ocean warming and deoxygenation are altering marine ecosystems, potentially resulting in a centuries-long, irreversible loss of habitable ocean volume in the upper 1000 m (Santana-Falcón et al., 2023). A

rapid growth in ocean heat content leads to considerable changes in sea ice melting, …"

7. L74-76: I think it's worth noting that there are some single model sets of simulations in existence either under constant concentrations (Dittus et al., 2024; Fabiano et al., 2023) or net zero emissions (King et al., 2024) but these are with a single simulation for a given forcing. This study is unusual in having an initial conditions ensemble for a given scenario under net zero emissions which is a
nice selling point of your paper.

A: We thank the reviewer for sharing this comment. We have added the following sentences in the revised manuscript:

"Regarding multi-centennial timescales, several stabilization experiments have been conducted, including those with constant atmospheric greenhouse gas concentrations (Dittus et al., 2024; Fabiano et al., 2024) or net-zero $CO_2$ emission simulations (King et al., 2024), to study climate projections under stabilized warming and the dependence on different levels of forcing. However, these experiments are based on single simulations. For this study, we have chosen to investigate forced changes in the climate system out to the year 2500 by extending 10 members of the 100-member Community Earth System Model 2 large ensemble….."

8. L104-105: Could you clarify whether the simulations are run in emissions or concentration-driven mode? I'm assuming the extensions beyond 2100 are emissions driven but it's not as clear as it could be. It's also not clear if the SSP3-7.0 simulations are emissions or concentration driven. You might interested in Sanderson et al., (2024) that discusses the merits of emissions-driven simulations.

A: We thank the reviewer for pointing this out. Our simulations are concentration-driven, meaning the greenhouse gas concentrations (defined as dry air mole fractions) from historical observations and the SSP3-7.0 scenario were prescribed throughout the entire simulation period (e.g., for $CO_2$, atmospheric $CO_2$ mole fraction is specified directly from 1850 to 2500). We have clarified this point in Section 2. We also added the study that the reviewer pointed out:

"Meinshausen et al. (2020) provided greenhouse gas concentrations, defined as dry air mole fractions, for both standard and extended SSP scenarios. They used the reduced-complexity climate-carbon-cycle model MAGICC7.0 ('Model for the Assessment of Greenhouse Gas Induced Climate Change') to produce future greenhouse gas concentrations driven by harmonized SSP greenhouse gas emissions (Gidden et al., 2019) and extended emissions beyond 2100. To extend the CESM2-LE from 2101 to 2500, we followed the extended SSP3-7.0 protocol, a concentration-driven configuration. In this extended scenario, fossil and industrial $CO_2$ emissions are effectively ramped down to zero by 2250 (Meinshausen et al., 2020), as shown in Fig. 1a. Figure 1b presents the time evolution of global mean greenhouse gas mole fractions ($CO_2$, $CH_4$, $N_2O$, and CFCs)

which are prescribed in these simulations under the historical (1850-2014), standard SSP3-7.0

(2015-2100) and extended SSP3-7.0 (2101-2500) scenario forcings. The global mean atmospheric

$CO_2$ mole fraction at the end of the 25th century, as provided by the extended SSP3-7.0 scenario, is approximately 1371 ppm."

9. Figure 1c,d: I assume the red lines are observations? In c it doesn't look like you're plotting anomalies from the observational period but the caption suggests you are.

A: We have modified the caption of Fig. 1 as follows:

"Figure 1: Time series of global mean (a) fossil fuel and industrial $CO_2$ emissions and (b)

greenhouse gas mole fractions over 1850-2500 for the CESM2-LE extension simulations. Values are taken from Meinshausen et al. (2020). Time series of global fields over 1850-2500 for 10

ensemble members for (c) top-of-atmosphere radiative imbalance (W m$^{-2}$) along with the CERES-

EBAF product (red) (Loeb et al., 2018; Loeb et al., 2009) and (d) anomalies of global mean surface air temperature (°C) along with HadCRUT4 (red) (Morice et al., 2012). In (c) and (d), bold lines represent ensemble means, and dark and light shading represent 1 standard deviation (SD) and 2

SD variability. In (d), observed and simulated temperature anomalies are calculated with respect to the period spanned by the observations (period 1950-2019).

10. L149-153: Could you clarify if this is annual-average sea ice extent? Of course, there are strong seasonal cycles and you are likely reaching "ice free" conditions at some times of the year. I'm wondering if this might be contributing to the decreased ensemble spread shown in Figure 2.

A: In Fig. 2, we considered the annually averaged fields, as clarified in the figure caption in the revised manuscript. Figure R1 shows the sea ice extent for the Arctic and Southern Ocean. The

Arctic sea ice extent in March (maximum) is projected to decline sharply after 2100, with reduced ensemble spread after the mid-22nd century. This indicates that the sea ice decline will be pronounced across all ensemble members, contributing to the decrease in ensemble spread of the annual mean sea ice extent after the mid-22nd century. Additionally, the Arctic sea ice extent in

September (minimum) shows little variation after the mid-21st century, with nearly all ensemble members predicting ice-free conditions, further reducing the ensemble spread. Similarly, in the

Southern Ocean, the transition to ice-free conditions in March across all ensembles after 2100 may also contribute to the reduced ensemble spread in the annual mean sea ice extent after the 21st century, relative to the period 1850-2100.

[Figure]

**Figure R1: Sea ice extent (10$^6$ km$^2$) for the (a) Arctic and (b) Southern Ocean. Bold lines represent ensemble**
**means, and shadings represent 2 standard deviation variability.**

11. L153-156: Could note that in ZECMIP the AMOC projections are highly model dependent and diverse (MacDougall et al., 2022).

A: We have added the following sentence following the reviewer's suggestion:

"However, it is noted that in the idealized experiments of the Zero Emission Commitment Model

Intercomparison Project (ZECMIP), some models indicate AMOC strengthening, while others predict a continued decline after 50 years of CO$_2$ emissions cessation, illustrating the model dependency of future AMOC projections (MacDougall et al., 2022)."

12. L208-210: These are certainly alarming amounts of global and local warming which should be highlighted. I think it is worth noting though that CESM2 has quite a high ECS (Gettelman et al.,

2019).

A: We thank the reviewer for the suggestion. We have addressed this point and added the suggested reference (line 113 in the submitted version of the manuscript).

13. L222-239: It could be noted that the precipitation changes are likely to be quite model dependent but increases at high latitudes in Southern Hemisphere are likely also associated with the warming in the region (Grose & King, 2023).

A: We have added the text as follows:

"The projected precipitation changes over the southern mid-latitudes and the Southern Ocean are expected to be linked to both Southern Ocean warming and the resulting meridional temperature gradient reduction between the tropics and the Southern Ocean (Grose and King, 2023)."

14. Figure 4h: Title is a bit confusing because I think this is precipitation variability? In general figure 4 could be improved as some axis labels are missing and some colour bar labels may lead to misinterpretation.

A: The title in Fig. 4h is "Niño3.4 precipitation variability". Additionally, following the reviewer's comment, we have modified Fig. 4:

[Figure]

**Figure 4: Time-averaged across-ensemble standard deviation of the boreal winter (DJF) mean (a) surface**
**temperature (°C) and (e) precipitation (mm day⁻¹) over the period 1850-1949. Future changes of the standard**
**deviation values of (b, c) surface temperature and (f, g) precipitation over the period 2150-2249 and over the**
**period 2401-2500, respectively, relative to 1850-1949. These calculations involve first determining the standard**
**deviation across all ensemble members for the same time period, followed by averaging across time. (d, h) 30-**
**year running standard deviations of monthly Niño3.4 SST and precipitation, respectively.**

15. L269-285: The projected changes in MJO are indeed very interesting. I would caution that the
interpretation could be a bit more understated given this is a single model result. Certainly this
finding should motivate related analysis with other models.

A: We thank the reviewer for their comment here. We understand the reviewer's concerns
regarding the interpretation based on results from a single model. In response, we have added the
following sentences at the end of Section 3.1 in the revised manuscript:

"Although our study provides valuable insights, the results should be interpreted with caution due
to the limitations of using output from a single Earth system model to represent long-term changes
in climate variability. Therefore, further investigations utilizing simulations from multiple Earth
system models would be highly beneficial in identifying areas of general consistency across
models, as well as potential areas of substantial disagreement."

16. L310-312: I can see why this sentence is included but it reads like a non-sequitur, and I think
is unnecessary.

A: We have deleted the sentence in the revised manuscript.

17. L375: ZECMIP studies with a carbon cycle focus may be worth citing here too (e.g
MacDougall et al., 2020).

A: We have added the appropriate reference to this study. We thank the reviewer for this
suggestion.

18. L389-390: This answers my earlier question. This is a fairly large caveat to some of the analysis
which should be noted earlier in the Data and Methods section.

A: As mentioned earlier, our simulations are concentration-driven. We have clarified this point in
Section 2. We kindly suggest that the reviewer refer to our response to comment #8 for further
clarification.

19. Figure 10: Given the same colour scale is used throughout it would look better to have a single
larger colour bar.

A: Thank you for the suggestion. We have modified Fig. 10 in the revised manuscript.

[Figure]

**Figure 10: The changes in the future period 2150-2249 (first column) and period 2401-2500 (second column) relative to the historical period 1850-1949. (a, b) PO$_4$ (mmol m$^{-3}$), (c, d) NO$_3$ (mmol m$^{-3}$), (e, f) Fe (mmol m$^{-3}$), and (g, h) SiO$_3$ (mmol m$^{-3}$). The changes were calculated as the ratio of the future to historical period. Blue (<1.0) and red (>1.0) indicate future decreases and increases, respectively, relative to the historical period. All values are averaged over the upper 100 m depth.**

20. L555-562: Similarly to a previous comment, the wording here is a bit over-confident given the results are derived from a single model.

A: We thank the reviewer for pointing this out. We have made efforts to avoid making overly strong or forceful statements throughout the text. Additionally, in the last paragraph of Section 4 Summary and Discussion, we have added the following sentences:

"The Earth System is currently on a Hothouse Earth trajectory driven by human-induced greenhouse gas emissions (Steffen et al., 2018). Our simulations project climate states that are well outside the range of what humans have thus far experienced, making it challenging to present the astonishingly broad range of large impacts under sustained anthropogenic perturbations. We emphasize that our study aims to provide a comprehensive overview of these important changes. Given the considerable uncertainty in future decisions regarding anthropogenic emissions, we do not intend to suggest that the climate changes projected in our simulations are likely pathways for the evolution of climate states. However, our simulations may serve as a crucial illustration of a potential future that we should strive to avoid. A novel aspect of our simulations lies in the presentation of not only mean state changes but also of shifts in the variability of the Earth system across diverse temporal and spatial scales under the influence of strong anthropogenic forcing. Furthermore, we anticipate that the ensemble framework analyses presented here can contribute a foundational step toward understanding the mechanisms that can impact multi-century climate-carbon feedbacks. Further complementary simulations, as well as emission-driven scenarios (Sanderson et al., 2024), are needed to explore these potential feedbacks more thoroughly. Although the future climate projections are……"

**References**

Dittus, A. J., Collins, M., Sutton, R., and Hawkins, E.: Reversal of Projected European Summer Precipitation Decline in a Stabilizing Climate, Geophysical Research Letters, 51, e2023GL107448, https://doi.org/10.1029/2023GL107448, 2024.

Fabiano, F., Davini, P., Meccia, V. L., Zappa, G., Bellucci, A., Lembo, V., Bellomo, K., and Corti, S.: Multi-centennial evolution of the climate response and deep-ocean heat uptake in a set of abrupt stabilization scenarios with EC-Earth3, Earth Syst. Dynam., 15, 527-546, 10.5194/esd-15-527-2024, 2024.

Gidden, M. J., Riahi, K., Smith, S. J., Fujimori, S., Luderer, G., Kriegler, E., van Vuuren, D. P., van den Berg, M., Feng, L., Klein, D., Calvin, K., Doelman, J. C., Frank, S., Fricko, O., Harmsen, M., Hasegawa, T., Havlik, P., Hilaire, J., Hoesly, R., Horing, J., Popp, A., Stehfest, E., and Takahashi, K.: Global emissions pathways under different socioeconomic scenarios for use in CMIP6: a dataset of harmonized emissions trajectories through the end of the century, Geosci. Model Dev., 12, 1443-1475, 10.5194/gmd-12-1443-2019, 2019.

Grose, M. R. and King, A. D.: The circulation and rainfall response in the southern hemisphere extra-tropics to climate stabilisation, Weather and Climate Extremes, 41, 100577, 2023.

King, A. D., Ziehn, T., Chamberlain, M., Borowiak, A. R., Brown, J. R., Cassidy, L., Dittus, A. J., Grose, M., Maher, N., Paik, S., Perkins-Kirkpatrick, S. E., and Sengupta, A.: Exploring climate stabilisation at different global warming levels in ACCESS-ESM-1.5, Earth Syst. Dynam., 15, 1353-1383, 10.5194/esd-15-1353-2024, 2024.

Loeb, N. G., Wielicki, B. A., Doelling, D. R., Smith, G. L., Keyes, D. F., Kato, S., Manalo-Smith, N., and Wong, T.: Toward optimal closure of the Earth's top-of-atmosphere radiation budget, Journal of Climate, 22, 748-766, 2009.

Loeb, N. G., Doelling, D. R., Wang, H., Su, W., Nguyen, C., Corbett, J. G., Liang, L., Mitrescu, C., Rose, F. G., and
Kato, S.: Clouds and the earth's radiant energy system (CERES) energy balanced and filled (EBAF) top-of-
atmosphere (TOA) edition-4.0 data product, Journal of climate, 31, 895-918, 2018.

MacDougall, A. H., Mallett, J., Hohn, D., and Mengis, N.: Substantial regional climate change expected following
cessation of $CO_2$ emissions, Environmental Research Letters, 17, 114046, 2022.

Meinshausen, M., Nicholls, Z. R. J., Lewis, J., Gidden, M. J., Vogel, E., Freund, M., Beyerle, U., Gessner, C., Nauels,
A., Bauer, N., Canadell, J. G., Daniel, J. S., John, A., Krummel, P. B., Luderer, G., Meinshausen, N., Montzka,
S. A., Rayner, P. J., Reimann, S., Smith, S. J., van den Berg, M., Velders, G. J. M., Vollmer, M. K., and Wang,
R. H. J.: The shared socio-economic pathway (SSP) greenhouse gas concentrations and their extensions to
2500, Geosci. Model Dev., 13, 3571-3605, 10.5194/gmd-13-3571-2020, 2020.

Morice, C. P., Kennedy, J. J., Rayner, N. A., and Jones, P. D.: Quantifying uncertainties in global and regional
temperature change using an ensemble of observational estimates: The HadCRUT4 data set, Journal of
Geophysical Research: Atmospheres, 117, 2012.

van Vuuren, D., O'Neill, B., Tebaldi, C., Chini, L., Friedlingstein, P., Hasegawa, T., Riahi, K., Sanderson, B.,
Govindasamy, B., and Bauer, N.: The Scenario Model Intercomparison Project for CMIP7 (ScenarioMIP-
CMIP7), EGUsphere, 2025, 1-38, 2025.